# EFFICIENT EXPLORATION THROUGH BAYESIAN DEEP Q-NETWORKS

## ABSTRACT

We propose Bayesian Deep Q-Networks (BDQN), a principled and a practical Deep Reinforcement Learning (DRL) algorithm for Markov decision processes (MDP). It combines Thompson sampling with deep-$Q$ networks (DQN). Thompson sampling ensures efficient exploration-exploitation trade-off in high dimensions. It is typically carried out through posterior sampling over the model parameters, which makes it computationally expensive. To overcome this limitation, we directly incorporate uncertainty over the value ($Q$) function. Further, we only introduce randomness in the last layer (i.e. the output layer) of the DQN and use independent Gaussian priors on the weights. This allows us to efficiently carry out Thompson sampling through Gaussian sampling and Bayesian Linear Regression (BLR), which has fast closed-form updates. The rest of the layers of the $Q$ network are trained through back propagation, as in a standard DQN. We apply our method to a wide range of Atari games in Arcade Learning Environments and compare BDQN to a powerful baseline: the double deep Q-network (DDQN). Since BDQN carries out more efficient exploration, it is able to reach higher rewards substantially faster: in less than 5M±1M interactions for almost half of the games to reach DDQN scores while a typical run of DDQN is 50-200M. We also establish theoretical guarantees for the special case when the feature representation is $d$-dimensional and fixed. We show that the Bayesian regret of posterior sampling RL (PSRL) and frequentist regret of the optimism in the face of uncertainly (OFU) after $N$ time step are upper bounded by $\widetilde{\mathcal{O}}(d\sqrt{N})$ which are tight up-to logarithmic factors. To the best of our knowledge, these are the first model free theoretical guarantee for continuous state-action space MDP, beyond the tabular setting.

## 1 INTRODUCTION

One of the central challenges in reinforcement learning (RL) is to design efficient exploration-exploitation trade-off that also scales to high-dimensional state and action spaces. Recently deep RL has shown good promise in being able scale to high-dimensional (continuous) spaces. These successes are mainly demonstrated in simulated domains where exploration is considered to be inexpensive and simple exploration strategies are deployed, e.g. $\varepsilon$-greedy which uniformly explores over all the actions with $\varepsilon$ probability. Such exploration strategies inherently inefficient for complex high-dimensional environments. On the other hand, more sophisticated strategies have mostly been limited to low dimensional MDPs. For example, OFU is only practical when the domain is small enough to be represented with lookup tables for the $Q$-values (Jaksch et al., 2010; Brafman & Tennenholtz, 2003).

An alternative to optimism-under-uncertainty is Thompson Sampling (TS), a general sampling and randomization approach (in both frequentist and Bayesian settings) (Thompson, 1933). Under the Bayesian framework, Thompson sampling maintains a prior distribution over the environmental models (i.e. models of reward and dynamics), and updating the posterior based on observations collected. An action is chosen from the posterior distribution, the belief, such that it maximizes the expected return. Thompson sampling has been observed to work significantly better than optimistic approaches in many low dimensional settings such as contextual bandits (Chapelle & Li, 2011), small MDPs (Osband et al., 2013) and also has strong theoretical bounds (Russo & Van Roy, 2014a;b; Agrawal & Goyal, 2012; Osband et al., 2013; Abbasi-Yadkori & Szepesvári, 2015).

In the MDP setting, (model-based) Thompson Sampling involves sampling the parameters of reward and dynamics models, performing MDP planning using the sampled model and then computing the policy (Strens, 2000; Osband et al., 2013; Osband & Van Roy, 2014b;a). However, the computational costs of posterior sampling and planning scales as cubic in the problem dimension, which makes it intractable for large MDPs. Therefore, some form of function approximation is required to scale Thompson Sampling to high dimensions, and this can be either of the model, the Q-value function, or the policy. To address this, Osband et al. (2014) introduced randomized least-squares value iteration (RLSVI) which combines linear value function approximation with Bayesian regression to directly sample the value-function weights from a distribution. The authors prove a regret bound for this approach in tabular MDPs. This has been extended to continuous spaces by Osband et al. (2016), where deep networks are used to approximate the $Q$ function. Through a bootstrapped-ensemble approach, several deep-Q network (DQN) models are trained in parallel to approximate the posterior distribution. Other works use the posterior distribution over the parameters of each node in the network and employ variational approximation (Lipton et al., 2016b) or use noisy networks (Fortunato et al., 2017). These approaches significantly increase the computation cost compared to the standard DQN. For instance, the bootstrapped-ensemble incurs a computation overhead that is linear in the number of bootstrap models. Moreover, despite principled design of the methods in the above works, they only produced modest gains over DQN in empirical studies.

**Contribution 1 – Design of BDQN:** We introduce Bayesian Deep Q-Network (BDQN), a Thompson-sampling algorithm for deep RL. It is a simple approach that extends randomized least-squares value iteration (Osband et al., 2014) to deep networks. We introduce stochasticity only in the last layer of the $Q$-network using independent Gaussian priors on the weights. This allows us to do efficient approximate Thompson sampling[1] using Bayesian linear regression (BLR), which has fast closed-form updates and sampling from the resulting posterior distribution. The rest of the $Q$-network is trained through usual back propagation.

**Contribution 2 – Strong empirical results for BDQN:** We test BDQN on a wide range of Atari games (Bellemare et al., 2013; Machado et al., 2017), and compare our results to a powerful baseline: Double DQN (DDQN) (Van Hasselt et al., 2016) a bias-reduced extension of DQN. BDQN and DDQN use the same network architecture, and follow the same target objective, and differ only in the way they select actions: DDQN uses $\varepsilon$-greedy exploration while BDQN performs (approximate) Thompson sampling.

We found that BDQN is able to reach much higher cumulative rewards, and also at a higher speed, compared to DDQN on all the tested games. We also found that BDQN can be trained with much higher learning rates compared to DDQN. This is intuitive since BDQN has better exploration strategy. The cumulative reward (score) for BDQN at the end of training improves by a median of $300\%$ with a maximum of $80K\%$ in these games. Also, BDQN has $300\% \pm 40\%$ (mean and standard deviation) improvement over these games on area under the performance measure. This can be considered as a surrogate for sample complexity and regret. Indeed, no single measure of performance provides a complete picture of an algorithm, and we present detailed experiments in Section 5.

In terms of computational cost, BDQN is only slightly more expensive compared to DQN and DDQN. For the DQN in Atari games, this is the cost of inverting a $512 \times 512$ matrix every 100,000 time steps, which is negligible. On the other hand, more sophisticated Bayesian RL techniques are significantly more expensive and have not lead to large gains over DQN and DDQN (Osband et al., 2016).

**Contribution 3 – Tight Bayesian and frequentist regret upper bounds for continuous MDPs:** We establish theoretical guarantees for the special case when the feature representation is fixed (i.e. all layers except the last), and not learnt. We consider episodic MDPs with continuous space of states and actions such that the $Q$-function is a linear function of a given $d$-dimensional feature map. We show that when PSRL and OFUare deployed, respectively, the Bayesian regret and frequentist regret after $N$ time step are upper bounded by $\widetilde{\mathcal{O}}(d\sqrt{N})$, which is shown to be tight up-to logarithmic factors. Since linear bandits are a special case of episodic continuous MDPs (with horizon length 1), it implies that our regret bounds are tight in the dimension $d$ and in the number of episodes. The Bayesian bound matches the Bayesian regret bound of linear bandits (Russo & Van Roy, 2014a) and the frequentist bound matches the frequentist regret bound for linear bandits (Abbasi-Yadkori

---

[1]BDQN approximates the posterior resulting in approximating Thompson sample. In the remaining, we state Thompson Sampling and its approximation as Thomson samples unless we specify.

et al., 2011). To the best of our knowledge, these are the first model free theoretical guarantee for continuous MDPs beyond the tabular setting.

Thus, our proposed approach has several desirable features – faster learning and better sample complexity due to targeted exploration, negligible computational overhead due to simplicity, significant improvement in experiments, and tight theoretical bounds.

## 2 THOMPSON SAMPLING VS $\varepsilon$-GREEDY AND BOLTZMANN EXPLORATION

In a value approximation RL algorithm, there are different ways to manage the exploration-exploitation trade-off. DQN uses a naive $\varepsilon$-greedy for exploration, where with $\varepsilon$ probability it chooses a random action and with $1 - \varepsilon$ probability it chooses the greedy action based on the estimated $Q$ function. Note that there are only point estimates of the $Q$ function in DQN. In contrast, our proposed Bayesian approach BDQN maintains uncertainties over the estimated $Q$ function, and employs it to carry out Thompson Sampling based exploration. Here, we demonstrate the fundamental benefits of Thompson Sampling over $\varepsilon$-greedy and Boltzmann exploration strategies using simple examples. In Table 1, we list the three strategies and their properties.

$\varepsilon$-greedy is the simplest exploration strategy since it is uniformly random over all the non-greedy actions. Boltzmann exploration is an intermediate strategy since it uses the estimated $Q$ function to sample the actions. However, it does not maintain uncertainties over the $Q$ function estimation. In contrast, Thompson sampling also incorporates uncertainties over $Q$ estimation and utilizes most information for exploration strategy.

Consider the example in Figure 1(a) with current estimates and uncertainties of the $Q$ function over different actions. $\varepsilon$-greedy is wasteful since it assigns uniform probability to explore over 5 and 6, which are obviously sub-optimal when the uncertainty estimates are available. In this setting, a possible remedy is Boltzmann exploration since it assigns lower probability to actions 5 and 6 but randomizes with almost the same probabilities over the remaining actions.

Table 1: Characteristics of Thompson Sampling, $\varepsilon$-greedy, and Boltzmann exploration what information they use for exploration

| Strategy | Greedy-Action | Estimated $Q$-values | Estimated uncertainties |
|---|---|---|---|
| $\varepsilon$-greedy | ✓ | ✗ | ✗ |
| Boltzmann exploration | ✓ | ✓ | ✗ |
| Thompson Sampling | ✓ | ✓ | ✓ |

However, Boltzmann exploration is sub-optimal in settings where there is high variances. For example if the current $Q$ estimate is according to Figure 1(b), then Boltzmann exploration assigns almost equal probability to actions 5 and 6, even though action 6 has much higher uncertainty and needs to be explored more. Thus, both $\varepsilon$-greedy and Boltzmann exploration strategies are sub-optimal since they do not maintain an uncertainty estimate over the $Q$ function estimation. In contrast, Thompson sampling uses both estimated $Q$ function and its uncertainty estimates to carry out the most efficient exploration.

## 3 BAYESIAN DEEP Q-NETWORKS

Consider an MDP $M$ as a tuple $\langle \mathcal{X}, \mathcal{A}, P, P_0, R, \gamma \rangle$, with state space $\mathcal{X}$, action space $\mathcal{A}$, transition kernel $P$, initial state distribution $P_0$, accompanied with reward function of $R$, and discount factor $0 \leq \gamma < 1$. In value based model free RL, the core of most prominent approaches is to learn the Q-function through minimizing the Bellman residual (Lagoudakis & Parr, 2003; Antos et al., 2008)

$$\mathcal{L}(Q) = \mathbb{E}_\pi \left[ (Q(x, a) - r - \gamma Q(x', \hat{a}))^2 \right] \qquad (1)$$

and temporal difference (TD) update (Tesauro, 1995) where the tuple $(x, a, r, x')$ consists of a consecutive experiences under a behavior policy $\pi$. Mnih et al. (2015) carries the same idea, and propose DQN where the Q-function is parameterized by a deep network and $\hat{a} = \arg\max_{a'} Q(x', a')$. In order to reduce the bias of the estimator, DQN utilizes a target network $Q^{target}$, target value

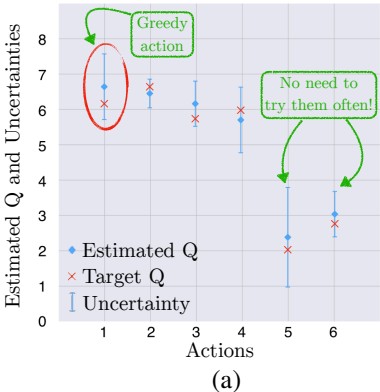 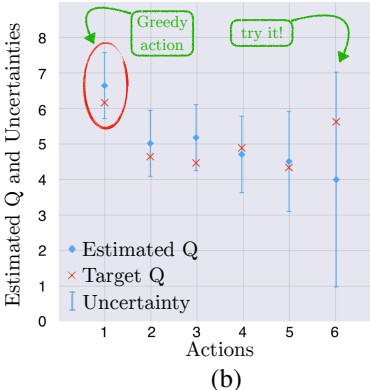

Figure 1: Thompson Sampling vs $\varepsilon$-greedy and Boltzmann exploration. $(a)$ $\varepsilon$-greedy is wasteful since it assigns uniform probability to explore over 5 and 6, which are obviously sub-optimal when the uncertainty estimates are available. Boltzmann exploration randomizes over actions even if the optimal action is identifies. (b) Boltzmann exploration does not incorporate uncertainties over the estimated action-values and chooses actions 5 and 6 with similar probabilities while action 6 is significantly more uncertain. Thomson Sampling is a simple remedy to all these issues.

---

**Algorithm 1** BDQN

---

1: Initialize $\theta$, $\theta^{target}$, $w_a$, $w_a^{target}$, $Cov_a$ $\forall a$, and $B$
2: **for** episode = 1,2,3... **do**
3:     **for** $t = 1$ to the end of episode **do**
4:         Draw sample $w_a \sim \mathcal{N}\left(w_a^{target}, Cov_a\right)$ $\forall a$ every $T^{sample}$
5:         Set $\theta^{target} \leftarrow \theta$ every $T^{target}$
6:         Update $w_a^{target}$ and $Cov_a$, $\forall a$ using $B$ experiences every $T^{Bayes\ target}$
7:         Execute $a_t = \arg\max_{a'} w_{a'}^\top \phi_\theta(x_t)$, observe reward $r_t$, successor State $x_{t+1}$
8:         Store transition $(x_t, a_t, r_t, x_{t+1})$ in the replay buffer
9:         Sample a random minibatch of transitions $(x_\tau, a_\tau, r_\tau, x_{\tau+1})$ from replay buffer
10:        $y_\tau \leftarrow \begin{cases} r_\tau & \text{terminal } x_{\tau+1} \\ r_\tau + w_{\hat{a}}^{target\top} \phi_{\theta^{target}}(x_{\tau+1}), \ \hat{a} := \arg\max_{a'} w_{a'}^\top \phi_\theta(x_{\tau+1}) & \text{non-terminal } x_{\tau+1} \end{cases}$
11:        $\theta \leftarrow \theta - \eta \cdot \nabla_\theta (y_\tau - w_{a_\tau}^\top \phi_\theta(x_\tau))^2$

---

$y = r + \gamma Q^{target}(x', \hat{a})$, and approaches the regression on the empirical estimates of the loss $\mathcal{L}(Q, Q^{target})$;

$$\mathcal{L}(Q, Q^{target}) = \mathbb{E}_\pi \left[ (Q(x, a) - y)^2 \right] \tag{2}$$

i.e., $\widehat{\mathcal{L}}(Q, Q^{target})$. A DQN agent, once in a while updates the $Q^{target}$ network and sets it to the $Q$ network, follows the regression in Eq.2 with the new target value and provides a biased estimator of the target. To mitigate the bias in this estimator, Van Hasselt et al. (2016) proposes DDQN and instead use $\hat{a} = \arg\max_{a'} Q^{target}(x', a')$. We deploy this approach for the rest of this paper.

DQN architecture consists of a deep neural network where the Q-function is approximated as a linear function of the feature representation layer $\phi_\theta(x) \in \mathbb{R}^d$ parametrized by $\theta$, i.e., for all pairs of state-action $\forall a \in \mathcal{A}, x \in \mathcal{X}$, we have $Q(x, a) = \phi_\theta(x)^\top w_a$ with $w_a \in \mathcal{R}^d$, the output layer. Consequently, the target model has the same architecture as the $Q$, and consists of $\phi^{\theta^{target}}(\cdot) \in \mathbb{R}^d$, the feature representation of target network, and $w^{target}_a$, $\forall a \in \mathcal{A}$ the target weight. For a given tuple of experiences $(x, a, r, x')$, and $\hat{a} = \arg\max_{a'} \phi^{\theta^{target}\top} w_{a'}^{target}$

$$Q(x, a) = \phi_\theta(x)^\top w_a \rightarrow y := r + \gamma \phi^{\theta^{target}}(x') w^{target}_{\hat{a}}$$

The regression in Eq. 2 induces linear regression in the learning of the output layer, i.e., $w_a$'s. In this work, we utilize the DQN architecture and instead propose to use BLR (Rasmussen & Williams,

2006) in learning of the output layer. Through BLR, we efficiently approximate the distribution over the Q-values, capture the uncertainty over the Q-fucntion estimation, and design a efficient exploration and exploitation strategy using Thompson Sampling.

By deploying BLR on the feature representation layer, we approximate the posterior distribution of each $w_a$, resulting in the posterior distribution of the $Q$-function. As in BLR methods, we maintain a Gaussian prior $\mathcal{N}(0, \sigma^2 I)$ with the target value $y \sim w_a^\top \phi_\theta(x) + \epsilon$ for each weight vector where $\epsilon \sim \mathcal{N}(0, \sigma_\epsilon^2)$ is an i.i.d. Gaussian noise.

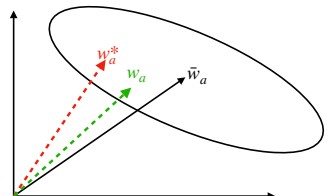

Given a experience replay buffer $\mathcal{D} = \{x_\tau, a_\tau, y_\tau\}_{\tau=1}^D$, we construct $|\mathcal{A}|$ (number of actions) disjoint datasets $\mathcal{D}_a$ for each action with $a_\tau = a$. For each action $a$, we construct a matrix $\Phi_a^\theta \in \mathbb{R}^{d \times |\mathcal{D}_a|}$, the concatenation of feature vectors $\{\phi_\theta(x_i)\}_{i=1}^{|\mathcal{D}_a|}$, and $\mathbf{y}_a \in \mathbb{R}^{|\mathcal{D}_a|}$, the concatenation of target values in set $\mathcal{D}_a$. Finally, we approximate the posterior distribution of $w_a$ as follows:

Figure 2: BDQN deploys Thompson Sampling to, sample $w_a \; \forall a \in A$ around the empirical mean $\overline{w}_a$ with $w_a^*$ the underlying parameter of interest.

$$w_a \sim \mathcal{N}\left(\overline{w}_a, Cov_a\right), \;\; \overline{w}_a := \frac{1}{\sigma_\epsilon^2} Cov_a \Phi_a^\theta \mathbf{y}_a, \;\; Cov_a := \left(\frac{1}{\sigma_\epsilon^2}\Phi_a^\theta \Phi_a^{\theta^\top} + \frac{1}{\sigma^2}I\right)^{-1} \quad (3)$$

Fig. 2 expresses that the covariance matrix induces an ellipsoid around the estimated mean of the approximated posterior and samples drawn through Thompson Sampling are mainly close to this mean. A sample of $Q(x,a)$ is $w_a^\top \phi_\theta(x)$ where $w_a$ is drawn from the posterior distribution Fig. 2. In BDQN, every $T^{sample}$ times step, we draw a new $w_a, \; \forall a \in \mathcal{A}$ and follow the resulting policy, i.e., $a_{\text{TS}} := \max_a w_a^\top \phi_\theta(x)$. We simultaneously train the feature network under the loss $(y_\tau - w_{a_\tau}^\top \phi_\theta(x_\tau))^2$ with $x_\tau, a_\tau, y_\tau$ experiences from the replay buffer i.e.

$$\theta \leftarrow \theta - \eta \cdot \nabla_\theta (y_\tau - \left[w_{a_\tau}^\top \phi_\theta(x_\tau)\right]_{a_\tau})^2 \quad (4)$$

We update the target network every $T^{target}$ steps and set $\theta^{target}$ to $\theta$. With the period of $T^{Bayes\; target}$, we update the posterior distribution using a minibatch of $B$ randomly chosen experiences in the replay buffer, and set the $w_a^{target} = \overline{w}_a, \; \forall a \in \mathcal{A}$ which is the mean of the posterior distribution( more details in Section A.5. We describe BDQN algorithm in Alg. 1.

## 4 BAYESIAN REGRET BOUND

| **Algorithm 2** PSRL |
|---|
| 1: Input: the prior and likelihood |
| 2: **for** episode t= 1,2,... **do** |
| 3:      $\omega_t \sim$ posterior distribution |
| 4:      **for** $h = 0$ to the end of episode **do** |
| 5:          Follow $\pi_t$ policy induced by $\omega_t$ |
| 6:          Update the posterior |

| **Algorithm 3** OFU |
|---|
| 1: Input: $\sigma$, $\lambda$ and $\delta$ |
| 2: **for** episode = 1,2,... **do** |
| 3:      choose optimistic $\widetilde{\pi}_t$ in $\mathcal{C}_{t-1}(\delta)$ |
| 4:      **for** $h = 1$ to the end of episode **do** |
| 5:          Follow $\widetilde{\pi}_t$ policy |
| 6:          Update the confidence $\mathcal{C}_t(\delta)$ |

In this section we provide the analysis of Bayesian regret upper bound of PSRL Alg. 2 and frequentis regret upper bound of optimism Alg. 3 when the feature representation is given and fixed. Consider a finite horizon stochastic MDP $M := \langle \mathcal{X}, \mathcal{A}, P, P_0, R, \gamma, H \rangle$, an MDP with horizon length $H$ and $0 \le \gamma \le 1$. We consider a class of MDPs where the optimal $Q$-function is a linear transformation of $\phi(\cdot, \cdot) := \mathcal{X} \times \mathcal{A} \to \mathcal{R}^d$, i.e., $Q_{\pi^*}^{\omega^*}(x,a) := \phi(x,a)^\top \omega^*$, $\forall x, a \in \mathcal{X} \times \mathcal{A}$, where $\omega^* \in \mathbb{R}^d$ and $\pi^* : \mathcal{X} \to \mathcal{A}$ as $\pi^*(x) := \arg\max_{a \in \mathcal{A}} Q_{\pi^*}^{\omega^*}(x,a)$. Let $V_{\pi^*}^{\omega^*}$ denote the corresponding value function. The following is the generative model of the environment;

$$\overline{\mathbf{R}} := \Psi\mathbf{R} = \Phi(x^{\{1:H\}}, a^{\{1:H\}})\omega^* + \Psi\nu(x^{\{1:H\}}, a^{\{1:H\}}), \; where, \; \Psi := \begin{bmatrix} 1 & \gamma & \gamma^2 & \dots & \gamma^{H-1} \\ 0 & 1 & \gamma & \dots & \gamma^{H-2} \\ \vdots & \dots & \dots & \dots & \vdots \\ 0 & \dots & \dots & \dots & 1 \end{bmatrix}$$

The vector $\mathbf{R} \in \mathbb{R}^H$ is the random vector of rewards in an episode and $\overline{\mathbf{R}} \in \mathbb{R}^H$ the corresponding per step return. $x^{\{1:H\}}, a^{\{1:H\}}$ is the sequence of states and actions, the matrix $\mathbf{\Phi}(x^{\{1:H\}}, a^{\{1:H\}}) \in \mathbb{R}^{H \times d}$ is row-wise concatenation of their features and $\nu(x^{\{1:H\}}, a^{\{1:H\}}) \in \mathbb{R}^d$ the noise. Alg. 2 maintains a prior over the vector $\omega^*$ and updating the posterior over time. At the beginning of an episode $t$, the agent draws $\omega_t$ from the posterior, and follows its induced policy $\pi_t$ where $a := \arg\max_{a \in \mathcal{A}} \phi^\top(x, a)\omega_t, \forall x \in \mathcal{X}$. Alg. 3, at the beginning of $t$'th episode, exploits the so-far collected samples and estimates $\omega^*$ up to a high probability confidence interval $C_{t-1}$ i.e., $\omega^* \in C_t$ with high probability. At each time step $h$, given a state $x_t^h$, the agent follows the optimistic policy; $\widetilde{\pi}_t(x_t^h) = \arg\max_{a \in \mathcal{A}} \max_{\omega \in \mathcal{C}_{t-1}} \phi^\top(X_t^h, a)\omega$. Through exploration and exploitation, we show that the confidence set $\mathcal{C}_t$ shrinks with the rate of $\widetilde{\mathcal{O}}\left(1/\sqrt{t}\right)$ resulting in less and less per step regret (Lemma 1 in Appendix B). Similar to the linear bandit (Abbasi-Yadkori et al., 2011), consider the following generic assumptions,

- The noise vector $\nu$ is sub-Gaussian vector with parameter $\sigma$. (Assumption 1 in Appendix B)

- $\|\omega^*\|_2 \le L_\omega$ and $trace\left(\mathbf{\Phi}(\cdot)\mathbf{\Phi}(\cdot)^\top\right) \le L$, a.s.

- Maximum expected cumulative reward of an episode condition on the states of that episode is in $[0, 1]$.

For any prior and likelihood satisfying the mentioned assumptions, we have;

**Theorem 1** (Bayesian Regret). *For an episodic MDP with episode length $H$, $\gamma = 1$, feature mapping $\phi(x, a) \in \mathbb{R}^d$, the Thompson sampling on $\omega$, Alg. 2, after $T$ episodes, guarantees;*

$$\textbf{\textit{BayesReg}}_T := \mathbb{E}\left[\sum_t^T \left[V_{\pi^*}^{\omega^*} - V_{\widetilde{\pi}_t}^{\omega^*}\right]\right] = \mathcal{O}\left(d\sqrt{TH}\log(T)\right)$$

*Proof is Appendix B.2.* Note that $N = TH$ is the number of agent-environment interactions.

**Theorem 2** (Frequentist Regret). *For an episodic MDP with episode length $H$, $\gamma = 1$, feature mapping $\phi(x, a) \in \mathbb{R}^d$, the optimism on $\omega$, Alg. 3, after $T$ episodes, guarantees;*

$$\textbf{\textit{Reg}}_T := \mathbb{E}\left[\sum_t^T \left[V_{\pi^*}^{\omega^*} - V_{\widetilde{\pi}_t}^{\omega^*}\right] | \omega^*\right] = \mathcal{O}\left(d\sqrt{TH}\log(T)\right)$$

*Proof is given in the Appendix B.1.*

**Remark 1.** *For any discount factor $\gamma$, $0 \le \gamma \le 1$, both theorems 1 and 2 hold if we replace $\sqrt{H}$ with a smaller constant $\|[1, \gamma, \gamma^2, \ldots, \gamma^{H-1}]\|_2$ in the theorem statements (see Section B.3).*

These bounds are similar to those in linear bandits (Abbasi-Yadkori et al., 2011; Russo & Van Roy, 2014a) and linear quadratic control (Abbasi-Yadkori & Szepesvári, 2011), i.e. $\widetilde{\mathcal{O}}(d\sqrt{T})$. Since for $H = 1$, this problem reduces to linear bandit and for linear bandit the lower bound is $\Omega(d\sqrt{T})$ therefore, our bound is order optimal in $d$ and $T$.

## 5 EXPERIMENTS

We apply BDQN on a variety of Atari games using the Arcade Learning Environment (Bellemare et al., 2013) through OpenAI Gym[2] (Brockman et al., 2016). As a baseline, we run the DDQN algorithm and evaluate BDQN on the measures of sample complexity and score. All the implementations are coded in MXNet framework (Chen et al., 2015). The details on architecture, Appendix A.1, learning rate Appendix A.3, computation A.4. In Appendix A.2 we describe how we spend less than two days on a single game for the hyper parameter choices which is another evidence of significance of BDQN.

---

[2]Each input frame is a pixel-max of the two consecutive frames. We detailed the environment setting in the implementation code

Table 2: Comparison of scores and sample complexities (scores in the first two columns are average of 100 consecutive episodes). The scores of DDQN$^+$ are the reported scores of DDQN in Van Hasselt et al. (2016) after running it for 200M interactions at evaluation time where the $\varepsilon = 0.001$. Bootstrap DQN (Osband et al., 2016), CTS, Pixel, Reactor (Ostrovski et al., 2017) are borrowed from the original papers. For NoisyNet (Fortunato et al., 2017), the scores of NoisyDQN are reported. Sample complexity, $SC$: the number of samples the BDQN requires to beat the human score (Mnih et al., 2015)(" $-$ " means BDQN could not beat human score). $SC^+$: the number of interactions the BDQN requires to beat the score of DDQN$^+$.

| Game | BDQN | DDQN | DDQN$^+$ | Bootstrap | NoisyNet | CTS | Pixel | Reactor | Human | SC | $SC^+$ | Step |
|------|------|------|---------|-----------|----------|-----|-------|---------|-------|------|--------|------|
| Amidar | **5.52k** | 0.99k | 0.7k | 1.27k | 1.5k | 1.03k | 0.62k | 1.18k | 1.7k | 22.9M | 4.4M | 100M |
| Alien | 3k | 2.9k | 2.9k | 2.44k | 2.9k | 1.9k | 1.7k | **3.5k** | 6.9k | - | 36.27M | 100M |
| Assault | **8.84k** | 2.23k | 5.02k | 8.05k | 3.1k | 2.88k | 1.25k | 3.5k | 1.5k | 1.6M | 24.3M | 100M |
| Asteroids | **14.1k** | 0.56k | 0.93k | 1.03k | 2.1k | 3.95k | 0.9k | 1.75k | 13.1k | 58.2M | 9.7M | 100M |
| Asterix | **58.4k** | 11k | 15.15k | 19.7k | 11.0 | 9.55k | 1.4k | 6.2k | 8.5k | 3.6M | 5.7M | 100M |
| BeamRider | 8.7k | 4.2k | 7.6k | **23.4k** | 14.7k | 7.0k | 3k | 3.8k | 5.8k | 4.0M | 8.1M | 70M |
| BattleZone | **65.2k** | 23.2k | 24.7k | 36.7k | 11.9k | 7.97k | 10k | 45k | 38k | 25.1M | 14.9M | 50M |
| Atlantis | 3.24M | 39.7k | 64.76k | 99.4k | 7.9k | 1.8M | 40k | **9.5M** | 29k | 3.3M | 5.1M | 40M |
| DemonAttack | 11.1k | 3.8k | 9.7k | 82.6k | 26.7k | **39.3k** | 1.3k | 7k | 3.4k | 2.0M | 19.9M | 40M |
| Centipede | **7.3k** | 6.4k | 4.1k | 4.55k | 3.35k | 5.4k | 1.8k | 3.5k | 12k | - | 4.2M | 40M |
| BankHeist | 0.72k | 0.34k | 0.72k | **1.21k** | 0.64k | 1.3k | 0.42k | 1.1k | 0.72k | 2.1M | 10.1M | 40M |
| CrazyClimber | 124k | 84k | 102k | **138k** | 121k | 112.9k | 75k | 119k | 35.4k | 0.12M | 2.1M | 40M |
| ChopCmd$^3$ | **72.5k** | 0.5k | 4.6k | 4.1k | 5.3k | 5.1k | 2.5k | 4.8k | 9.9k | 4.4M | 2.2M | 40M |
| Enduro | 1.12k | 0.38k | 0.32k | 1.59k | 0.91k | 0.69k | 0.19k | **2.49k** | 0.31k | 0.82M | 0.8M | 30M |
| Pong | **21** | 18.82 | **21** | 20.9 | **21** | 20.8 | 17 | 20 | 9.3 | 1.2M | 2.4M | 5M |

**Baselines:**   We implemented DDQN and BDQN exactly the same way as described in Van Hasselt et al. (2016). We also attempted to implement a few other deep RL methods that employ strategic exploration, e.g., (Osband et al., 2016; Bellemare et al., 2016). Unfortunately we encountered several implementation challenges where neither code nor the implementation details was publicly available. Despite the motivation of this work on sample complexity, since we do not have access to the performance plots of these methods, the least is to report their final scores. To try to illustrate the performance of our approach we instead, extracted the best reported scores from a number of state-of-the-art deep RL methods and include them in Table 2, which is the only way to bring a comparison. We compare against DDQN, as well as DDQN$^+$ which is the reported scores of DDQN in Van Hasselt et al. (2016) at evaluation time where the $\varepsilon = 0.001$. Furthermore, we compared against scores of Bootstrap DQN (Osband et al., 2016), NoisyNet (Fortunato et al., 2017), CTS, Pixel, Reactor (Ostrovski et al., 2017) which are borrowed from the original papers. For NoisyNet, the scores of NoisyDQN are reported. We also provided the sample complexity, $SC$: the number of interactions BDQN requires to beat the human score (Mnih et al., 2015)(" $-$ " means BDQN could not beat human score) and $SC^+$: the number of interactions the BDQN requires to beat the score of DDQN$^+$. Note that these are not perfect comparisons, as there are additional details that are not included in the papers, i.e. it is hard to just compare the reported results (an issue that has been discussed extensively recently, e.g. (Henderson et al., 2017)).[4]. Moreover, when the regret analysis of an algorithm is considered, no evaluation phase required, and the reported results of BDQN are those while exploration. It is worth noting that, the scores during evaluation are much higher than those during the exploration and exploitation period, Appendix A.8.

**Results:**   The results are provided in Fig. 3. We observe that BDQN significantly improve the sample complexity of DDQN and reaches the highest score of DDQN in much fewer number of interactions than DDQN needs. We expected BDQN, due to it better exploration-exploitation strategy, to improve the regret and enhance the sample complexity, but we also observed a significantly improvement in scores. It worth noting that since BDQN is designed to minimize the regret, and the study in Fig. 3 also for sample complexity analysis, either of the reported BDQN and DDQN scores are while

---

[4]To further reproducibility, we released our codes and trained models. Since DRL experiments are expensive, we also have released the recorded arrays of returns. We also implemented bootstrapped DQN (Osband et al., 2016) and released the code but we were not able to reproduce their results beyond the performance of random policy

exploration. For example, DDQN gives score of 18.82 during the learning phase, but setting $\varepsilon$ to zero, it mostly gives the score of 21. In addition to the Table 2, we also provided the score ratio as well area under the performance plot ratio comparisons in Table 3.

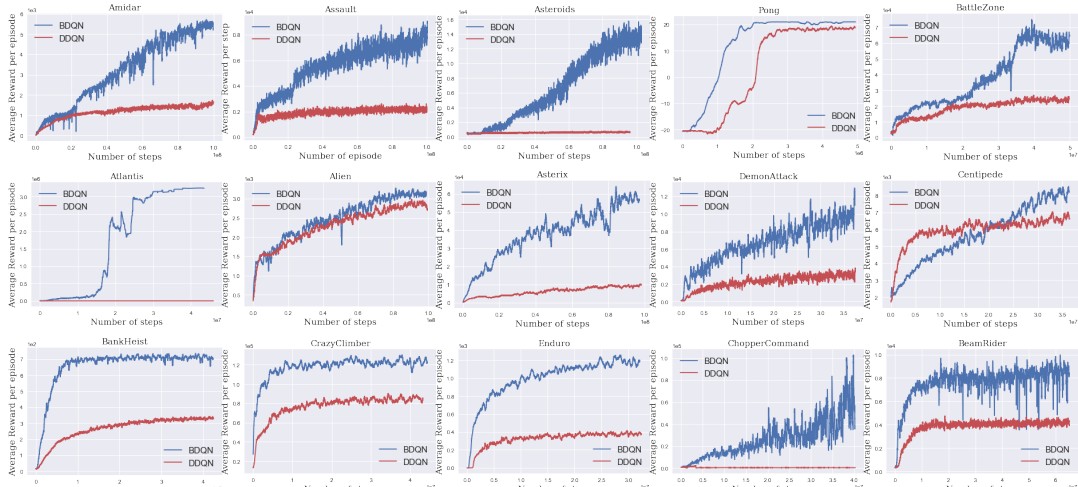

Figure 3: The comparison between DDQN and BDQN

For the game *Atlantis*, DDQN[+] gives score of $64.67k$ during the evaluation phase, while BDQN reaches score of $3.24M$ after $20M$ interactions. As it is been shown in Fig. 3, BDQN saturates for *Atlantis* after 20M interactions. We realized that BDQN reaches the internal *OpenAIGym* limit of $max\_episode$, where relaxing it improves score after $15M$ steps to $62M$, Appendix A.7. We observe that BDQN immediately learns significantly better policies due to its efficient explore/exploit in a much shorter period of time. Since BDQN on game *Atlantis* promise a big jump around time step $20M$, we ran it five more times in order to make sure it was not just a coincidence Appendix A.7 Fig. 7. For the game Pong, we ran the experiment for a longer period but just plotted the beginning of it in order to observe the difference. Due to cost of deep RL methods, for some games, we run the experiment until a plateau is reached.

## 6  RELATED WORK

The complexity of the exploration-exploitation trade-off has been deeply investigated in RL literature for both continuous and discrete MDPs (Kearns & Singh, 2002; Brafman & Tennenholtz, 2003; Asmuth et al., 2009; Kakade et al., 2003; Ortner & Ryabko, 2012). Jaksch et al. (2010) investigates the regret analysis of MDPs with finite state and action where Optimism in Face of Uncertainty (OFU) principle is deployed to guarantee a regret upper bound, while Ortner & Ryabko (2012) relaxes it to a continuous state space and propose a sublinear regret bound. Azizzadenesheli et al. (2016a) deploys OFUand propose a regret upper bound for Partially Observable MDPs (POMDPs) using spectral methods (Anandkumar et al., 2014). Furthermore, Bartók et al. (2014) tackles a general case of partial monitoring games and provides minimax regret guarantee. For linear quadratic models OFU is deployed to provide an optimal regret bound (Abbasi-Yadkori & Szepesvári, 2011).

In multi-arm bandit, there are compelling empirical pieces of evidence that Thompson Sampling sometimes provides better results than optimism-under-uncertainty approaches (Chapelle & Li, 2011), while also the performance guarantees are preserved (Russo & Van Roy, 2014a; Agrawal & Goyal, 2012). A natural adaptation of this algorithm to RL, posterior sampling RL (PSRL) Strens (2000) also shown to have good frequentist and Bayesian performance guarantees (Osband et al., 2013; Abbasi-Yadkori & Szepesvári, 2015).

Even though the theoretical RL addresses the exploration and exploitation trade-offs, these problems are still prominent in empirical reinforcement learning research (Mnih et al., 2015; Abel et al., 2016; Azizzadenesheli et al., 2016b). On the empirical side, the recent success in the video games has sparked a flurry of research interest. Following the success of Deep RL on Atari games (Mnih et al., 2015) and the board game Go (Silver et al., 2017), many researchers have begun exploring practical

applications of deep reinforcement learning (DRL). Some investigated applications include, robotics (Levine et al., 2016), self-driving cars (Shalev-Shwartz et al., 2016), and safety (Lipton et al., 2016a). Inevitably for PSRL, the act of posterior sampling for policy or value is computationally intractable with large systems, so PSRL can not be easily leveraged to high dimensional problems (Ghavamzadeh et al., 2015; Engel et al., 2003; Dearden et al., 1998; Tziortziotis et al., 2013). To remedy these failings Osband et al. (2017) consider the use of randomized value functions. For finite state-action space MDP, Osband et al. (2014) propose posterior sampling directly on the space of Q-functions and provide a strong Bayesian regret bound guarantee. To approximate the posterior, they use BLR on one-hot encoding of state-action and improve the computation complexity of PSRL. The approach deployed in BDQN is strongly related and similar to this work, and is a generalization to continues state-action space MDPs.

To combat the computational and scalability shortcomings, Osband et al. (2016) suggests a bootstrapped-ensemble approach that trains several models in parallel to approximate the posterior distribution. Other works suggest using a variational approximation to the Q-networks (Lipton et al., 2016b) or a concurrent work on noisy network (Fortunato et al., 2017). However, most of these approaches significantly increase the computational cost of DQN and neither approach produced much beyond modest gains on Atari games. Interestingly, the Bayesian approach as a technique for learning a neural network has been deployed for object recognition and image caption generation where its significant advantage has been verified Snoek et al. (2015).

Concurrently, Levine et al. (2017) proposes least squares temporal difference which learns a linear model on the feature representation in order to estimate the $Q$-function while $\varepsilon$-greedy exploration is employed and improvement on 5 tested Atari games is provided. Out of these 5 games, one is common with our set of 15 games which BDQN outperform it by factor of $360\%$ (w.r.t. the score reported in their paper). As motivated by theoretical understanding, our empirical study shows that performing Bayesian regression instead, and sampling from the result, can yield a substantial benefit, indicating that it is not just the higher data efficiency at the last layer, but that leveraging an explicit uncertainty representation over the value function is of substantial benefit.

Drop-out, as another randomized exploration method is proposed by Gal & Ghahramani (2016) but Osband et al. (2016) investigates the sufficiency of the estimated uncertainty and hardness in driving suitable exploitation out of it. We also implemented the dropout version of DDQN and compared its performance against BDQN, DDQN, DDQN$^+$, and the random policy (the policy which chooses actions uniformly at random) but did not observe much gain beyond the performance of the random policy (See section A.6). As stated before, in spite of the novelties proposed by the methods, mentioned in this section, neither of them, including TSbased approaches, produced much beyond modest gains on Atari games while BDQN provides significant improvements in terms of both sample complexity and final performance.

# 7   CONCLUSION

In this work we proposed BDQN, a practical Thompson sampling based RL algorithm which provides efficient exploration/exploitation in a computationally efficient manner. It involved making simple modifications to the DDQN architecture by replacing the last layer with Bayesian linear regression. Under the Gaussian prior, we obtained fast closed-form updates for the posterior distribution. We demonstrated significantly faster training and much better performance in many games compared to the reported results of a wide number of state-of-the-art baselines. We also established theoretical guarantees for an episodic MDP with continuous state and action spaces in the special case where the feature representation is fixed. We derived an order-optimal frequentist and Bayesian regret bound of $\widetilde{\mathcal{O}}(d\sqrt{N})$ after $N$ time steps.

In future, we plan to extend the analysis to the more general frequentist setting. The current theoretical guarantees are mainly developed for the class of linear functions. For general class of functions, optimism in the face of uncertainty is deployed to guarantee a tight probably approximately correct (PAC) bound (Jiang et al., 2016) but the proposed algorithm requires solving a NP-hard problem at each iteration. We aim to further provide a tight theoretical bound through Thomson sampling while also preserving computational feasibility.

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

# A    APPENDIX

In the main text, for simplicity, we use the terms "BLR" for i.i.d. samples and BLR for non i.i.d. samples exchangeable, even though, technically, the do not have equal meaning. In RL, the data is not i.i.d, and we extend the BLR to non i.i.d. setting by deploying additional Martingale type argument and handles the data with temporal dependency.

Table 3: $1st$ column: score ratio of BDQN to DDQN run for same number of time steps. $2nd$ column: score ratio of BDQN to DDQN$^+$. $3rd$ column: score ratio of BDQN to human scores reported at Mnih et al. (2015). $4th$ column: Area under the performance plot ration (AuPPr) of BDQN to DDQN. AuPPr is the integral of area under the performance plot ration. For Pong, since the scores start form $-21$, we shift it up by 21. $5th$ column: Sample complexity, $SC$: the number of samples the BDQN requires to beat the human score (Mnih et al., 2015)(" $-$ " means BDQN could not beat human score). $6th$ column: $SC^+$: the number of samples the BDQN requires to beat the score of DDQN$^+$. We run both BDQN and DDQN for the same number of times steps, stated in the last column.

| Game | $\dfrac{\text{BDQN}}{\text{DDQN}}$ | $\dfrac{\text{BDQN}}{\text{DDQN}^+}$ | $\dfrac{\text{BDQN}}{\text{HUMAN}}$ | AuPPr | SC | $SC^+$ | Steps |
|---|---|---|---|---|---|---|---|
| Amidar | 558% | 788% | 325% | 280% | 22.9M | 4.4M | 100M |
| Alien | 103% | 103% | 43% | 110% | - | 36.27M | 100M |
| Assault | 396% | 176% | 589% | 290% | 1.6M | 24.3M | 100M |
| Asteroids | 2517% | 1516% | 108% | 680% | 58.2M | 9.7M | 100M |
| Asterix | 531% | 385% | 687% | 590% | 3.6M | 5.7M | 100M |
| BeamRider | 207% | 114% | 150% | 210% | 4.0M | 8.1M | 70M |
| BattleZone | 281% | 253% | 172% | 180% | 25.1M | 14.9M | 50M |
| Atlantis | 80604% | 49413% | 11172% | 380% | 3.3M | 5.1M | 40M |
| DemonAttack | 292% | 114% | 326% | 310% | 2.0M | 19.9M | 40M |
| Centipede | 114% | 178% | 61% | 105% | - | 4.2M | 40M |
| BankHeist | 211% | 100% | 100% | 250% | 2.1M | 10.1M | 40M |
| CrazyClimber | 148% | 122% | 350% | 150% | 0.12M | 2.1M | 40M |
| ChopperCommand | 14500% | 1576% | 732% | 270% | 4.4M | 2.2M | 40M |
| Enduro | 295% | 350% | 361% | 300% | 0.82M | 0.8M | 30M |
| Pong | 112% | 100% | 226% | 130% | 1.2M | 2.4M | 5M |

## A.1    NETWORK ARCHITECTURE:

The input to the network part of BDQN is $4 \times 84 \times 84$ tensor with a rescaled and averaged over channels of the last four observations. The first convolution layer has 32 filters of size 8 with a stride of 4. The second convolution layer has 64 filters of size 4 with stride 2. The last convolution layer has 64 filters of size 3 followed by a fully connected layer with size 512. We add a BLR layer on top of this.

## A.2    CHOICE OF HYPER-PARAMETERS:

For BDQN, we set the values of $W^{target}$ to the mean of the posterior distribution over the weights of BLR with covariances $Cov$ and draw $W$ from this posterior. For the fixed $W$ and $W^{target}$, we randomly initialize the parameters of network part of BDQN, $\theta$, and train it using RMSProp, with learning rate of $0.0025$, and a momentum of $0.95$, inspired by (Mnih et al., 2015) where the discount factor is $\gamma = 0.99$, the number of steps between target updates $T^{target} = 10k$ steps, and weights $W$ are re-sampled from their posterior distribution every $T^{sample}$ steps. We update the network part of BDQN every 4 steps by uniformly at random sampling a mini-batch of size 32 samples from the replay buffer. We update the posterior distribution of the weight set $W$ every $T^{Bayes\ target}$ using mini-batch of size $B$ (if the size of replay buffer is less than $B$ at the current step, we choose the minimum of these two ), with entries sampled uniformly form replay buffer. The experience replay contains the $1M$ most recent transitions. Further hyper-parameters are equivalent to ones in DQN setting.

For the BLR, we have noise variance $\sigma_\epsilon$, variance of prior over weights $\sigma$, sample size $B$, posterior update period $T^{Bayes\ target}$, and the posterior sampling period $T^{sample}$. To optimize for this set of hyper-parameters we set up a very simple, fast, and cheap hyper-parameter tuning procedure which proves the robustness of BDQN. To find the first three, we set up a simple hyper-parameter search. We used a pretrained DQN model for the game of *Assault*, and removed the last fully connected layer in order to have access to its already trained feature representation. Then we tried combination of $B = \{T^{target}, 10 \cdot T^{target}\}$, $\sigma = \{1, 0.1, 0.001\}$, and $\sigma_\epsilon = \{1, 10\}$ and test for 1000 episode of the game. We set these parameters to their best $B = 10 \cdot T^{target}, \sigma = 0.001, \sigma = 1$.

The above hyper-parameter tuning is cheap and fast since it requires only a few times the $B$ number of forwarding passes. For the remaining parameters, we ran BDQN ( with weights randomly initialized) on the same game, *Assault*, for $5M$ time steps, with a set of $T^{Bayes\ target} = \{T^{target}, 10 \cdot T^{target}\}$ and $T^{sample} = \{\frac{T^{target}}{10}, \frac{T^{target}}{100}\}$, where BDQN performed better with choice of $T^{Bayes\ target} = 10 \cdot T^{target}$. For both choices of $T^{sample}$, it performs almost equal and we choose the higher one to reduce the computation cost. We started off with the learning rate of $0.0025$ and did not tune for that. Thanks to the efficient Thompson sampling exploration and closed form BLR, BDQN can learn a better policy in an even shorter period of time. In contrast, it is well known for DQN based methods that changing the learning rate causes a major degradation in the performance (Fig. 4). The proposed hyper-parameter search is very simple and an exhaustive hyper-parameter search is likely to provide even better performance.

## A.3 Learning rate:

It is well known that DQN and DDQN are sensitive to the learning rate and change of learning rate can degrade the performance to even worse than random policy. We tried the same learning rate as BDQN, $0.0025$, for DDQN and observed that its performance drops. Fig. 4 shows that the DDQN with higher learning rates learns as good as BDQN at the very beginning but it can not maintain the rate of improvement and degrade even worse than the original DDQN with learning rate of $0.00025$.

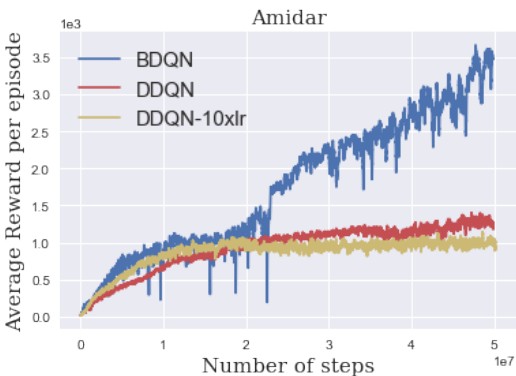

Figure 4: Effect of learning rate on DDQN

## A.4 Computational and sample cost comparison:

For a given period of game time, the number of the backward pass in both BDQN and DQN are the same where for BDQN it is cheaper since it has one layer (the last layer) less than DQN. In the sense of fairness in sample usage, for example in duration of $10 \cdot T^{Bayes\ target} = 100k$, all the layers of both BDQN and DQN, except the last layer, sees the same number of samples, but the last layer of BDQN sees 16 times fewer samples compared to the last layer of DQN. The last layer of DQN for a duration of $100k$, observes $25k = 100k/4$ (4 is back prob period) mini batches of size 32, which is $16 \cdot 100k$, where the last layer of BDQN just observes samples size of $B = 100k$. As it is mentioned in Alg. 1, to update the posterior distribution, BDQN draws $B$ samples from the replay buffer and needs to compute the feature vector of them. Therefore, during the $100k$ interactions for the learning procedure, DDQN does $32 * 25k$ of forward passes and $32 * 25k$ of backward passes, while BDQN

Table 4: The comparison of BDQN, DDQN, Dropout-DDQN and random policy. Dropout-DDQN as another randomization strategy provides a deficient estimation of uncertainty and results in poor exploration/exploitation trade-off.

| Game | BDQN | DDQN | DDQN$^+$ | Dropout-DDQN | Random Policy | Step |
|---|---|---|---|---|---|---|
| CrazyClimber | 124k | 84k | 102k | 19k | 11k | 40M |
| Atlantis | 3.24M | 39.7k | 64.76k | 7.7k | 12.85k | 40M |
| Enduro | 1.12k | 0.38k | 0.32k | 0.27k | 0 | 30M |
| Pong | 21 | 18.82 | 21 | -18 | -20.7 | 5M |

does same number of backward passes (cheaper since there is no backward pass for the final layer) and $36 * 25k$ of forward passes. One can easily relax it by parallelizing this step along the main body of BDQN or deploying on-line posterior update methods.

## A.5 THOMPSON SAMPLING FREQUENCY:

The choice of Thompson sampling update frequency can be crucial from domain to domain. Theoretically, we show that for episodic learning, the choice of sampling at the beginning of each episode, or a bounded number of episodes is desired. If one chooses $T^{sample}$ too short, then computed gradient for backpropagation of the feature representation is not going to be useful since the gradient get noisier and the loss function is changing too frequently. On the other hand, the network tries to find a feature representation which is suitable for a wide range of different weights of the last layer, results in improper waste of model capacity. If the Thompson sampling update frequency is too low, then it is far from being Thompson sampling and losses the randomized exploration property. We are interested in a choice of $T^{sample}$ which is in the order of upper bound on the average length of each episode of the Atari games. The current choice of $T^{sample}$ is suitable for a variety of Atari games since the length of each episode is in range of $\mathcal{O}(T^{sample})$ and is infrequent enough to make the feature representation robust to big changes.

For the RL problems with shorter a horizon we suggest to introduce two more parameters, $\tilde{T}^{sample}$ and each $\tilde{w}_a$ where $\tilde{T}^{sample}$, the period that of each $\tilde{w}_a$ is sampled out of posterior, is much smaller than $T^{sample}$ and $\tilde{w}_a, \forall a$ are used for Thompson sampling where $w_a, \forall a$ are used for backpropagation of feature representation. For game Assault, we tried using $\tilde{T}^{sample}$ and each $\tilde{w}_a$ but did not observe much a difference, and set them to $T^{sample}$ and each $w_a$. But for RL setting with a shorter horizon, we suggest using them.

## A.6 DROPOUT AS A RANDOMIZED EXPLORATION STRATEGY

Dropout, as another randomized exploration method, is proposed by Gal & Ghahramani (2016), but Osband et al. (2016) argue about the deficiency of the estimated uncertainty and hardness in driving a suitable exploration and exploitation trade-off from it (Appendix A in (Osband et al., 2016)). They argue that Gal & Ghahramani (2016) does not address the fundamental issue that for large networks trained to convergence all dropout samples may converge to every single datapoint. As also observed by (Dhillon et al., 2018), dropout might results in a ensemble of many models, but all almost the same (converge to the very same model behavior). We also implemented the dropout version of DDQN, Dropout-DDQN, and ran it on four randomly chosen Atari games (among those we ran for less than 50M time steps). We observed that the randomization in Dropout-DDQN is deficient and results in performances worse than DDQN on these four Atari games, Fig. 5. In Table 4 we compare the performance of BDQN, DDQN, DDQN$^+$, and Dropout-DDQN, as well as the performance of the random policy, borrowed from Mnih et al. (2015). We observe that the Dropout-DDQN not only does not outperform the plain $\varepsilon$-greedy DDQN, it also sometimes underperforms the random policy. For the game Pong, we also ran Dropout-DDQN for $50M$ time steps but its average performance did not get any better than -17. For the experimental study we used the default dropout rate of $0.5$ to mitigate its collapsing issue.

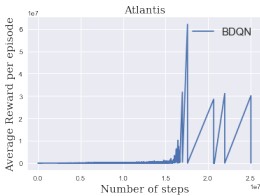

Figure 5: The comparison between DDQN, BDQN and Dropout-DDQN

## A.7 Further investigation on Atlantis:

After removing the maximum episode length limit for the game Atlantis, BDQN gets the score of 62M. This episode is long enough to fill half of the replay buffer and make the model perfect for the later part of the game but losing the crafted skill for the beginning of the game. We observe in Fig. 6 that after losing the game in a long episode, the agent forgets a bit of its skill and loses few games but wraps up immediately and gets to score of $30M$. To overcome this issue, one can expand the replay buffer size, stochastically store samples in the reply buffer where the later samples get stored with lowest chance, or train new models for the later parts of the episode. There are many possible cures for this interesting observation and while we are comparing against DDQN, we do not want to advance BDQN structure-wise.

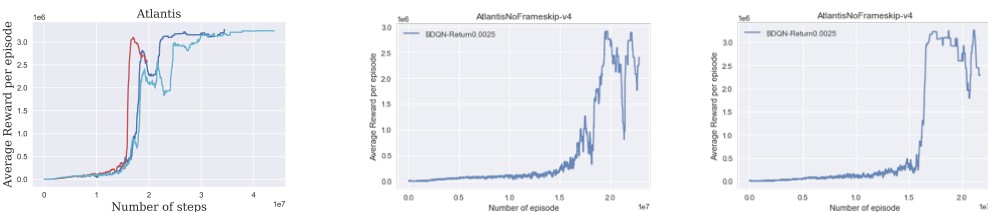

Figure 6: BDQN on Atlantis after removing the limit on max of episode length hits the score of $62M$ in $16M$ samples.

Figure 7: A couple of more runs of BDQN where the jump around $15M$ constantly happens

## A.8 Further discussion on Reproducibility

In Table 2, we provide the scores of bootstrap DQN (Osband et al., 2016) and NoisyNet[5]Fortunato et al. (2017) along with BDQN. These score are directly copied from their original papers and we did not make any change to them. We also desired to report the scores of count-based method (Ostrovski et al., 2017), but unfortunately there is no table of score in that paper in order to provide them here.

In order to make it easier for the readers to compare against the results in Ostrovski et al. (2017), we visually approximated their plotted curves for $CTS$, $Pixel$, and $Reactor$, and added them to the Table 2. We added these numbers just for the convenience of the readers. Surely we do not argue any scientific meaning for them and leave it to the readers to interpret them.

Table 2 shows a significant improvement of BDQN over these baselines. Despite the simplicity and negligible computation overhead of BDQN over DDQN, we can not scientifically claim that BDQN

---

[5]This work does not have scores of Noisy-net with DDQN objective function but it has Noisy-net with DQN objective which are the scores reported in Table 2

outperforms these baselines by just looking at the scores in Table2 because we are not aware of their detailed implementation as well as environment details. For example, in this work, we directly implemented DDQN by following the implementation details mentioned in the original DDQN paper and the scores of our DDQN implementation during the evaluation time almost matches the scores of DDQN reported in the original paper. But the reported scores of implemented DDQN in Osband et al. (2016) are much different from the reported score in the original DDQN paper.

### A.9  A SHORT DISCUSSION ON SAFETY

In BDQN, as mentioned in Eq. 3, the prior and likelihood are conjugate of each others. Therefore, we have a closed form posterior distribution of the discounted return, $\sum_{t=0}^{N} \gamma^t r_t | x_0 = x, a_0 = a, \mathcal{D}_a$, approximated as

$$\mathcal{N}\left(\frac{1}{\sigma_\epsilon^2}\phi^\theta(x)^\top \Xi_a \Phi_a^\theta \mathbf{y}_a, \phi^\theta(x)^\top \Xi_a \phi^\theta(x)\right)$$

One can use this distribution and come up with a safe RL criterion for the agent (García & Fernández, 2015). Consider the following example; for two actions with the same mean, if the estimated variance over the return increases, then the action becomes more unsafe Fig. 8. By just looking at the low and high probability events of returns under different actions we can approximate whether an action is safe to take.

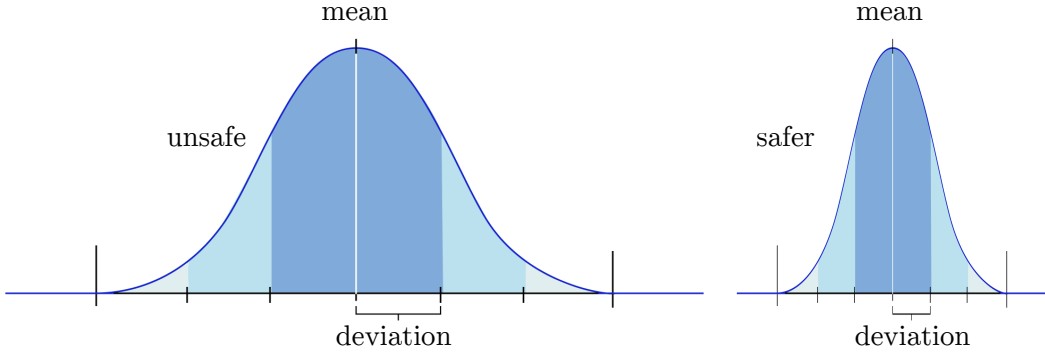

Figure 8: Two actions, with the same mean, but the one with higher variance on the return might be less safe than the one with narrower variance on the return.

## B  BAYESIAN AND FREQUENTIST REGRET, PROOF OF THEOREMS 1,2

**Modeling:**  We consider the following linear model over the $Q$ functions and returns for episodic MDPs with episode length of $H$. We consider the discount factor to be any number $0 \leq \gamma \leq 1$. Let $\omega^*$ denotes the parameter corresponding the underlying model. For a given policy $\pi$ and the time step $h_t$ in the episode, condition on $X^{h_0}, A^{h_0}$ we have $\sum_{h=h_0}^{H} R_h = \phi(X^{h_0}, A^{h_0})^\top \omega^* + \sum_{h=h_0}^{H} \nu_h$ with $\phi(X, A) \in \mathcal{R}^d$. The noise process is correlated, dependent on the agent policy, and is not mean zero unless under policy $\pi^*$.

$$\Psi \mathbf{R} = \Phi(X^{\{1:H\}}, A^{\{1:H\}})\omega^* + \Psi\nu(X^{\{1:H\}}, A^{\{1:H\}}), \; where, \; \Psi := \begin{bmatrix} 1 & \gamma & \gamma^2 & \dots & \gamma^{H-1} \\ 0 & 1 & \gamma & \dots & \gamma^{H-2} \\ 0 & 0 & 1 & \dots & \vdots \\ \vdots & \dots & \dots & \dots & \vdots \\ 0 & \dots & \dots & \dots & 1 \end{bmatrix} \quad (5)$$

The vector $\mathbf{R} \in \mathbb{R}^H$ is a random vector, as stack of rewards in a episode, i.e., $\mathbf{R}^h = r^h$. The matrix $\Phi(X^{\{1:H\}}, X^{\{1:H\}}) \in \mathbb{R}^{H \times d}$ is a row-wised concatenation of sequence of state-action features, $\Phi^h = \phi(X^h, A^h)$ with $\Phi_h$ denotes the $h$'th row.

In order to provide more intuition on the model, we can exploit the property of the matrix $\Psi$ where

$$
\Psi^{-1} = \begin{bmatrix} 1 & \text{-}\gamma & 0 & \dots & 0 \\ 0 & 1 & \text{-}\gamma & \dots & 0 \\ 0 & 0 & 1 & \dots & \vdots \\ \vdots & \dots & \dots & \dots & \vdots \\ 0 & \dots & \dots & \dots & 1 \end{bmatrix}
$$

and rewrite Eq. 5 as follows:

$$
\mathbf{R}_t = \Psi^{-1}\mathbf{\Phi}_t\omega + \nu_t \tag{6}
$$

We denote $\Psi' := {\Psi^{-1}}^\top$. We build our analysis based on the former notation. We consider the feature representing matrix and weights for any $X^{\{1:H\}}, A^{\{1:H\}}$ to satisfy;

$$
trace\left( \mathbf{\Phi}(X^{\{1:H\}}, A^{\{1:H\}})\mathbf{\Phi}(X^{\{1:H\}}, A^{\{1:H\}})^\top \right) \leq L \,,\ \|\omega\|_2 \leq L_\omega
$$

The vector $\nu(X^{\{1:H\}}, A^{\{1:H\}}) \in \mathbb{R}^H$ is a state-action and policy dependent noise vector. The MDP assumption is crucial for the analysis in this paper. The noise $\nu^h$ encodes the randomness in transition to state $X^h$, possible stochasticity in policy i.e., $A^h$, reward $R^h$. For any $h$, the noises after time step $h$, $\{\nu^h, \nu^{h+1}, \dots, \nu^H\}$, are conditionally independent from the noises before time step $h'$, $\{\nu^1, \nu^2, \dots, \nu^{h-1}\}$ given $(X^h, A^h)$. Moreover, since $\nu^h$ encodes the stochasticity in transition and reward at time step $h$, conditional distribution of $\nu^h | (X^h, A^h)$ is mean zero and independent $\{\nu^0, \dots \{\nu^{h-1}, \nu^{h+1}, \dots, \nu^H\}$ other of since $\phi(X^h, A^h)^\top \omega^*$ represent the true $Q$ value of $X^h, A^h$.

The agent policy at each episode is denoted as $\pi_t$. In the remaining, we show how to estimate $\omega^*$ through data collected by following $\pi_t$. We denote $\hat{\omega}_t$, the estimation of $\omega^*$. We show over time the estimation concentrates around $\omega^*$ under the desire matrix norm. We further show, we can deploy this analysis an construct two algorithms, one based on PSRL, and another Optimism OFUto guaranteed Bayesian and frequentist regret upper bounds respectively. The main body of the following analyses on concentration of measure is a matrix extension of contextual linear bandit analyses Abbasi-Yadkori et al. (2011); Chu et al. (2011); Li et al. (2010); Rusmevichientong & Tsitsiklis (2010); Dani et al. (2008); Russo & Van Roy (2014a) and self normalized processes (de la Pena et al., 2004). Our regret analysis extends prior analysis in frequentist and Bayesian regret literature Osband et al. (2013); Jaksch et al. (2010).

We restate the Eq. 5 in its abstract way;

$$
\overline{\mathbf{R}}_t := \Psi\mathbf{R}_t = \mathbf{\Phi}_t\omega + \overline{\nu}_t \,,\quad where\ \ \overline{\nu}_t := \Psi\nu_t \tag{7}
$$

We consider the discount factor to be any number $0 \leq \gamma \leq 1$ for the modeling and for the sake of analysis we consider $\gamma = 1$ from now on which we later argue the bounds also hold for any $0 \leq \gamma \leq 1$.

**Assumption 1** (Sub-Gaussian random vector (Hsu et al., 2012)). *The noise model at time t, conditioned on the filtration $\mathcal{F}_{t-1}$ at time $t$ is sub-Gaussian random vector, i.e. there exists a parameter $\sigma \geq 0$ such that $\forall \alpha \in \mathcal{R}^d$*

$$
\mathbb{E}\left[ \exp\left( \alpha^\top \overline{\nu}_t \right) \Big| \mathcal{F}_{t-1} \right] \leq \exp\left( \|\alpha\|_2 \sigma^2/2 \right)
$$

The filtration at time t also includes $X^{\{1:H\}}, A^{\{1:H\}}$. Asm. 1 also implies that $\mathbb{E}\left[ \overline{\nu}_t \Big| \mathcal{F}_t \right] = 0$ which means that after drawing the matrix $\mathbf{\Phi}_t$, the random variable $\overline{\nu}_t$ is means zero. A similar assumption on the noise model is assumed in prior analyses of linear bandit (Abbasi-Yadkori et al., 2011).

Furthermore, we consider the maximum expected cumulative reward condition on states of a episode is at most 1.

Defined the following quantities for self-normalized processes;

$$
S_t := \sum_i^t \mathbf{\Phi}_i^\top \overline{\nu}_i, \quad \chi_t := \sum_{i=0}^t \mathbf{\Phi}_i^\top \mathbf{\Phi}_i, \quad \overline{\chi}_t = \chi_t + \widetilde{\chi}
$$

where $\widetilde{\chi}$ is a ridge regularization matrix and usually is equal to $\lambda I$.

**Lemma 1** (Confidence ellipsoid for problem in Eq. 5). *For a matrix problem in Eq. 5, given a sequence of* $\{\Phi_i, \mathbf{R}_i\}_{i=1}^T$, *under the Asm. 1 and the following estimator for* $\widehat{\omega}_t$

$$\widehat{\omega}_t := \left( \sum_i^t \Phi_i^\top \Phi_i + \lambda I \right)^{-1} \sum_i^t \left( \Phi_i^\top \overline{\mathbf{R}}_i \right)$$

*with probability at least* $1 - \delta$

$$\|\widehat{\omega}_\tau - \omega^*\|_{\overline{\chi}_t} \le \theta_t(\delta) : \sigma \sqrt{2 \log{(1/\delta)} + d \log{(1 + tL^2/\lambda)}} + \lambda^{1/2} L_\omega$$

*for all* $t \le T$ *where* $\|\omega^*\|_2 \le L_\omega$ *and* $trace\left( \Phi(\cdot)\Phi(\cdot)^\top \right) \le L$, *and therefore the confidence set is*

$$\mathcal{C}_t(\delta) := \{\omega \in \mathbb{R}^d : \|\widehat{\omega}_t - \omega\|_{\overline{\chi}_t} \le \theta_t(\delta)\}$$

Let $\Theta_T$ denote the event that the confidence bounds in Lemma 1 holds.

**Lemma 2** (Determinant Lemma). *For a sequence* $\sum_t^T \log\left( 1 + \|\Phi_t\|_{\overline{\chi}_t^{-1}} \right)$ *we have*

$$\sum_t^T \log\left( 1 + \|\Phi_t\|_{\overline{\chi}_t^{-1}} \right) \le d \log(\lambda + TL^2/d) \tag{8}$$

**Regret Definition**    We defined $V_\pi^\omega$, the value of policy $\pi$ applied on model specified by $\omega$ in Eq. 5. We restate the definition of the regret and Bayesian regret.

$$\mathbf{Reg}_T := \mathbb{E}\left[ \sum_t^T \left[ V_{\pi^*}^{\omega^*} - V_{\pi_t}^{\omega^*} \right] | \omega^* \right]$$

Where $\pi_t$ is the agent policy at time $t$

When there is a prior over the $\omega^*$ the expected Bayesian regret might be the target of the study.

$$\mathbf{BayesReg}_T := \mathbb{E}\left[ \sum_t^T \left[ V_{\pi^*}^{\omega^*} - V_{\pi_t}^{\omega^*} \right] \right]$$

### B.1    Optimism: Regret bound of Alg. 3

Given samples by following the agent policy $\pi_t'$ for each episode $t' \le t$ we estimate the $\widehat{\omega}_t$ as follows;

$$\widehat{\omega}_t := \left( \sum_i^t \Phi_i^\top \Phi_i + \lambda I \right)^{-1} \sum_i^t \left( \Phi_i^\top \overline{\mathbf{R}}_i \right)$$

Lemma 1 states that under event $\Theta_T$, $\|\widehat{\omega}_t - \omega^*\|_{\overline{\chi}_t} \le \theta_t(\delta)$. For an observed state $X_t^1$, the optimistic policy is

$$\widetilde{\pi}_t(X_t^h) = \arg\max_{a \in \mathcal{A}} \max_{\omega \in \mathcal{C}_{t-1}(\delta)} \phi^\top(X_t^h, a)\omega$$

Furthermore, we define state and policy dependent optimistic parameter $\widetilde{\omega}_t(\pi)$ as follows;

$$\widetilde{\omega}_t(\pi) := arg \max_{\omega \in \mathcal{C}_{t-1}(\delta)} \phi(X_t^1, \pi(X_t^1))^\top \omega$$

For ease of notation we drop the state in the notation. Following OFU, Alg. 3, we set $\pi_t = \widetilde{\pi}_t$ denote the optimistic policy. By the definition we have

$$V_{\widetilde{\pi}_t}^{\widetilde{\omega}_t(\widetilde{\pi}_t)}(X_t^h) \ge V_{\pi^*}^{\widetilde{\omega}_t(\pi^*)}(X_t^h)$$

Therefore, under the event $\Theta_T$ we have

$$\mathbf{Reg}_T := \mathbb{E}\left[\sum_t^T \left[\underbrace{V_{\pi^*}^{\omega^*}(X_t^1) - V_{\widetilde{\pi}_t}^{\omega^*}(X_t^1)}_{\Delta_t^{h=1}}\right]|\omega^*\right]$$

$$\leq \mathbb{E}\left[\sum_t^T \left[V_{\widetilde{\pi}_t}^{\widetilde{\omega}_t(\widetilde{\pi}_t)}(X_t^1) - V_{\pi^*}^{\widetilde{\omega}_t(\pi^*)}(X_t^1) + V_{\pi^*}^{\omega^*}(X_t^1) - V_{\widetilde{\pi}_t}^{\omega^*}(X_t^1)\right]|\omega^*\right]$$

$$= \mathbb{E}\left[\sum_t^T \left[V_{\widetilde{\pi}_t}^{\widetilde{\omega}_t(\widetilde{\pi}_t)}(X_t^1) - V_{\widetilde{\pi}_t}^{\omega^*}(X_t^1) + \underbrace{V_{\pi^*}^{\omega^*}(X_t^1) - V_{\pi^*}^{\widetilde{\omega}_t(\pi^*)}(X_t^1)}_{\leq 0}\right]|\omega^*\right]$$

Resulting in

$$\mathbf{Reg}_T \leq \mathbb{E}\left[\sum_t^T \left[V_{\widetilde{\pi}_t}^{\widetilde{\omega}_t(\widetilde{\pi}_t)}(X_t^1) - V_{\widetilde{\pi}_t}^{\omega^*}(X_t^1)\right]|\omega^*\right]$$

Lets defined $V_\pi^\omega(X_t^1,\ldots,X_t^{h'};h:\pi')$ as the value function following policy $\pi$ for $h$ time step then switching to policy $\pi'$ on the model $\omega$ and observing $X_t^1,\ldots X_t^{h'}$.

$$\mathbf{Reg}_T \leq \mathbb{E}\left[\sum_t^T \left[V_{\widetilde{\pi}_t}^{\widetilde{\omega}_t(\widetilde{\pi}_t)}(X_t^1) - V_{\widetilde{\pi}_t}^{\omega^*}(X_t^1)\right]|\omega^*\right]$$

$$= \mathbb{E}\left[\sum_t^T \left[V_{\widetilde{\pi}_t}^{\widetilde{\omega}_t(\widetilde{\pi}_t)}(X_t^1) - V_{\widetilde{\pi}_t}^{\omega^*}(X_t^1;1:\pi^*) + V_{\widetilde{\pi}_t}^{\omega^*}(X_t^1;1:\pi^*) - V_{\widetilde{\pi}_t}^{\omega^*}(X_t^1)\right]|\omega^*\right]$$

Given the generative model in Eq.5 we have

$$\mathbf{Reg}_T \leq \mathbb{E}\left[\sum_t^T \left[V_{\widetilde{\pi}_t}^{\widetilde{\omega}_t(\widetilde{\pi}_t)}(X_t^1) - V_{\widetilde{\pi}_t}^{\omega^*}(X_t^1;1:\pi^*) + V_{\widetilde{\pi}_t}^{\omega^*}(X_t^1;1:\pi^*) - V_{\widetilde{\pi}_t}^{\omega^*}(X_t^1)\right]|\omega^*\right]$$

$$= \mathbb{E}\left[\sum_t^T \left[\phi(X_t^1,\widetilde{\pi}_t(X_t^1))^\top \widetilde{\omega}_t(\widetilde{\pi}_t) - \phi(X_t^1,\widetilde{\pi}_t(X_t^1))^\top \omega^* + V_{\widetilde{\pi}_t}^{\omega^*}(X_t^1;1:\pi^*) - V_{\widetilde{\pi}_t}^{\omega^*}(X_t^1)\right]|\omega^*\right]$$

$$= \mathbb{E}\left[\sum_t^T \left[\phi(X_t^1,\widetilde{\pi}_t(X_t^1))^\top (\widetilde{\omega}_t(\widetilde{\pi}_t) - \omega^*) + \underbrace{V_{\widetilde{\pi}_t}^{\omega^*}(X_t^1;1:\pi^*) - V_{\widetilde{\pi}_t}^{\omega^*}(X_t^1)}_{(\Delta_t^{h=2})}\right]|\omega^*\right]$$

For $\Delta_t^{h=2}$ we deploy the similar decomposition and upper bound as $\Delta_t^{h=1}$

$$\Delta_t^2 := V_{\widetilde{\pi}_t}^{\omega^*}(X_t^1;1:\pi^*) - V_{\widetilde{\pi}_t}^{\omega^*}(X_t^1)$$

Since for both of $V_{\widetilde{\pi}_t}^{\omega^*}(X_t^1;1:\pi^*)$ and $V_{\widetilde{\pi}_t}^{\omega^*}(X_t^1)$ we follow the same policy on the same model for 1 time step, the reward at the first time step has the same distribution, therefore we have;

$$\Delta_t^{h=2} = E\left[V_{\widetilde{\pi}_t}^{\omega^*}(X_t^2;1:\pi^*) - V_{\widetilde{\pi}_t}^{\omega^*}(X_t^2)|X_t^1,\omega^*\right]$$

rustling in

$$\Delta_t^{h=2} = E\left[V_{\widetilde{\pi}_t}^{\omega^*}(X_t^2;1:\pi^*) - V_{\widetilde{\pi}_t}^{\omega^*}(X_t^2)|X_t^1,\omega^*\right]$$

$$\leq \mathbb{E}\left[\phi(X_t^2,\widetilde{\pi}_t(X_t^2))^\top (\widetilde{\omega}_t(\widetilde{\pi}_t) - \omega^*) + \underbrace{V_{\widetilde{\pi}_t}^{\omega^*}(X_t^2;2:\pi^*) - V_{\widetilde{\pi}_t}^{\omega^*}(X_t^2)}_{(\Delta_t^{h=3})}|X_t^1,\omega^*\right]$$

Similarly we can defined $\Delta_t^{h=3}, \ldots \Delta_t^{h=H}$. Therefore;

$$
\begin{aligned}
\mathbf{Reg}_T &\leq \mathbb{E}\left[\sum_t^T \sum_h^H \phi(X_t^h, \widetilde{\pi}_t(X_t^h))^\top (\widetilde{\omega}_t(\widetilde{\pi}_t) - \omega^*) \,|\omega^*\right] \\
&= \mathbb{E}\left[\sum_t^T \mathbb{1}^\top \boldsymbol{\Phi}_t (\widetilde{\omega}_t(\widetilde{\pi}_t) - \omega^*) \,|\omega^*\right] \\
&= \mathbb{E}\left[\sum_t^T \mathbb{1}^\top \boldsymbol{\Phi}_t \overline{\chi}_t^{-1/2} \overline{\chi}_t^{1/2} (\widetilde{\omega}_t(\widetilde{\pi}_t) - \omega^*) \,|\omega^*\right] \\
&\leq \mathbb{E}\left[\sqrt{H} \sum_t^T \|\boldsymbol{\Phi}_t\|_{\overline{\chi}_t^{-1}} \|\widetilde{\omega}_t(\widetilde{\pi}_t) - \omega^*\|_{\overline{\chi}_t} \,|\omega^*\right] \\
&\leq \mathbb{E}\left[\sqrt{H} \sum_t^T \|\boldsymbol{\Phi}_t\|_{\overline{\chi}_t^{-1}} 2\theta_{t-1}(\delta) \,|\omega^*\right]
\end{aligned}
\tag{9}
$$

Since the maximum expected cumulative reward, condition on states of a episode is at most 1, we have;

$$
\begin{aligned}
\mathbf{Reg}_T &\leq \sqrt{H}\mathbb{E}\left[\sum_t^T \min\{\|\boldsymbol{\Phi}_t\|_{\overline{\chi}_t^{-1}} 2\theta_{t-1}(\delta), 1\} \,|\omega^*\right] \\
&\leq \sqrt{H}\mathbb{E}\left[\sum_t^T \sum_h^H 2\theta_{t-1}(\delta) \min\{\|\boldsymbol{\Phi}_t\|_{\overline{\chi}_t^{-1}}, 1\} \,|\omega^*\right]
\end{aligned}
$$

Moreover, at time $T$, we can use Jensen's inequality, exploit the fact that $\theta_t(\delta)$ is an increasing function of $t$ and have

$$
\begin{aligned}
\mathbf{Reg}_T &\leq 2\mathbb{E}\left[\sqrt{TH\theta_T(\delta)^2 \sum_t^T \min\{\|\boldsymbol{\Phi}_t\|_{\overline{\chi}_t^{-1}}, 1\}} \,|\omega^*\right] \\
&\leq 2\left(\sigma\sqrt{2\log(1/\delta) + d\log(1 + TL^2/\lambda)} + \lambda^{1/2}L_\omega\right) \mathbb{E}\left[\sqrt{TH\sum_t^T \min\{\|\boldsymbol{\Phi}_t\|_{\overline{\chi}_t^{-1}}, 1\}} \,|\omega^*\right]
\end{aligned}
\tag{10}
$$

Now, using the fact that for any scalar $\alpha$ such that $0 \leq \alpha \leq 1$, then $\alpha \leq 2\log(1 + \alpha)$ we can rewrite the latter part of Eq. 10

$$
\sqrt{\sum_t^T \min\{\|\boldsymbol{\Phi}_t\|_{\overline{\chi}_t^{-1}}, 1\}} \leq 2\sum_t^T \log\left(1 + \|\boldsymbol{\Phi}_t\|_{\overline{\chi}_t^{-1}}\right)
$$

By applying the Lemma 2 and substituting the RHS of Eq. 8 into Eq. 10, we get

$$
\mathbf{Reg}_T \leq 2\left(\sigma\sqrt{2\log(1/\delta) + d\log(1 + TL^2/\lambda)} + \lambda^{1/2}L_\omega\right)\sqrt{2THd\log(\lambda + TL^2/d)}
\tag{11}
$$

with probability at least $1 - \delta$. If we set $\delta = 1/T$ then the probability that the event $\Theta_T$ holds is $1 - 1/T$ and we get regret of at most the RHS of Eq. 11, otherwise with probability at most $1/T$ we get maximum regret of $T$, therefore

$$
\mathbf{Reg}_T \leq 1 + 2\left(\sigma\sqrt{2\log(T) + d\log(1 + TL^2/\lambda)} + \lambda^{1/2}L_\omega\right)\sqrt{2THd\log(\lambda + TL^2/d)}
$$

and theorem statement follows.

### B.2 BAYESIAN REGRET OF ALG. 2

The analysis developed in the previous section, up to some minor modification, e.g., change of strategy to PSRL, directly applies to Bayesian regret bound, with a farther expectation over models.

When there is a prior over the $\omega^*$ the expected Bayesian regret might be the target of the study.

$$\mathbf{BayesReg}_T := \mathbb{E}\left[\sum_t^T \left[V_{\pi^*}^{\omega^*} - V_{\pi_t}^{\omega^*}\right]\right]$$

$$\mathbb{E}\left[\sum_t^T \left[V_{\pi^*}^{\omega^*} - V_{\pi_t}^{\omega^*} | \mathcal{H}_t\right]\right]$$

Here $\mathcal{H}_t$ is a multivariate random sequence which indicates history at the beginning of episode $t$ and $\pi_t$ is the policy following PSRL. For the remaining $\pi_t$ denotes the PSRL policy. As it mentioned in the Alg. 2, at the beginning of an episode, we draw $\omega_t$ for the posterior and the corresponding policy is

$$\pi_t(X_t^h) := \arg\max_{a \in \mathcal{A}} \phi(X_t^h, a)^\top \omega_t$$

Condition on the history $\mathcal{H}_t$, samples by following the agent policy $\pi_{t'}'$ for each episode $t' \le t$, similar to previous section we estimate the $\widehat{\omega}_t$ as follows;

$$\widehat{\omega}_t := \left(\sum_i^t \mathbf{\Phi}_i^\top \mathbf{\Phi}_i + \lambda I\right)^{-1} \sum_i^t \left(\mathbf{\Phi}_i^\top \mathbf{R}_i\right)$$

Lemma 1 states that under event $\Theta_T$, $\|\widehat{\omega}_t - \omega^*\|_{\overline{X}_t} \le \theta_t(\delta)$. Conditioned on $\mathcal{H}_t$, the $\omega_t$ and $\omega^*$ are equally distributed, then we have

$$\mathbb{E}\left[V_{\pi^*}^{\widetilde{\omega}_t(\pi^*)} | \mathcal{H}_t\right] = \mathbb{E}\left[V_{\pi_t}^{\widetilde{\omega}_t(\pi_t)} | \mathcal{H}_t\right]$$

Therefore, under event $\Theta_T$ holds for all $\omega^*$ we have

$$\mathbf{BayesReg}_T := \sum_t^T \mathbb{E}\left[\underbrace{V_{\pi^*}^{\omega^*}(X_t^1) - V_{\pi_t}^{\omega^*}(X_t^1)}_{\Delta_t^{h=1}} | \mathcal{H}_t\right]$$

$$= \sum_t^T \mathbb{E}\left[V_{\pi_t}^{\widetilde{\omega}_t(\pi_t)}(X_t^1) - V_{\pi^*}^{\widetilde{\omega}_t(\pi^*)}(X_t^1) + V_{\pi^*}^{\omega^*}(X_t^1) - V_{\pi_t}^{\omega^*}(X_t^1) | \mathcal{H}_t\right]$$

$$= \sum_t^T \mathbb{E}\left[V_{\pi_t}^{\widetilde{\omega}_t(\pi_t)}(X_t^1) - V_{\pi_t}^{\omega^*}(X_t^1) + \underbrace{V_{\pi^*}^{\omega^*}(X_t^1) - V_{\pi^*}^{\widetilde{\omega}_t(\pi^*)}(X_t^1)}_{\le 0} | \mathcal{H}_t\right]$$

Resulting in

$$\mathbf{BayesReg}_T \le \sum_t^T \mathbb{E}\left[V_{\pi_t}^{\widetilde{\omega}_t(\pi_t)}(X_t^1) - V_{\pi_t}^{\omega^*}(X_t^1) | \mathcal{H}_t\right]$$

Similar to optimism and defining $V_\pi^\omega(X_t^1, \ldots, X_t^{h'}; h : \pi')$ accomponeid with Eq.5 we have;

$$\mathbf{BayesReg}_T \leq \mathbb{E}\left[\sum_t^T \left[V_{\pi_t}^{\widetilde{\omega}_t(\pi_t)}(X_t^1) - V_{\pi_t}^{\omega^*}(X_t^1)|\mathcal{H}_t\right]\right]$$

$$= \mathbb{E}\left[\sum_t^T \left[V_{\pi_t}^{\widetilde{\omega}_t(\pi_t)}(X_t^1) - V_{\pi_t}^{\omega^*}(X_t^1; 1 : \pi^*) + V_{\pi_t}^{\omega^*}(X_t^1; 1 : \pi^*) - V_{\pi_t}^{\omega^*}(X_t^1)|\mathcal{H}_t\right]\right]$$

$$\leq \mathbb{E}\left[\sum_t^T \left[\phi(X_t^1, \pi_t(X_t^1))^\top \widetilde{\omega}_t(\pi_t) - \phi(X_t^1, \pi_t(X_t^1))^\top \omega^* + V_{\pi_t}^{\omega^*}(X_t^1; 1 : \pi^*) - V_{\pi_t}^{\omega^*}(X_t^1)|\mathcal{H}_t\right]\right]$$

$$= \mathbb{E}\left[\sum_t^T \left[\phi(X_t^1, \pi_t(X_t^1))^\top (\widetilde{\omega}_t(\pi_t) - \omega^*) + \underbrace{V_{\pi_t}^{\omega^*}(X_t^1; 1 : \pi^*) - V_{\pi_t}^{\omega^*}(X_t^1)}_{(\Delta_t^{h=2})}|\mathcal{H}_t\right]\right]$$

For $\Delta_t^{h=2}$ we deploy the similar decomposition and upper bound as $\Delta_t^{h=1}$

$$\Delta_t^2 := V_{\pi_t}^{\omega^*}(X_t^1; 1 : \pi^*) - V_{\pi_t}^{\omega^*}(X_t^1)$$

Since for both of $V_{\pi_t}^{\omega^*}(X_t^1; 1 : \pi^*)$ and $V_{\pi_t}^{\omega^*}(X_t^1)$ we follow the same distribution over policies and models for 1 time step, the reward at the first time step has the same distribution, therefore we have;

$$\Delta_t^{h=2} = E\left[V_{\pi_t}^{\omega^*}(X_t^2; 1 : \pi^*) - V_{\pi_t}^{\omega^*}(X_t^2)|X_t^1, \mathcal{H}_t\right]$$

rustling in

$$\Delta_t^{h=2} = E\left[V_{\pi_t}^{\omega^*}(X_t^2; 1 : \pi^*) - V_{\pi_t}^{\omega^*}(X_t^2)|X_t^1, \mathcal{H}_t\right]$$

$$\leq \mathbb{E}\left[\phi(X_t^2, \pi_t(X_t^2))^\top (\widetilde{\omega}_t(\pi_t) - \omega^*) + \underbrace{V_{\pi_t}^{\omega^*}(X_t^2; 2 : \pi^*) - V_{\pi_t}^{\omega^*}(X_t^2)}_{(\Delta_t^{h=3})}|X_t^1, \mathcal{H}_t\right]$$

Similarly we can defined $\Delta_t^{h=3}, \ldots \Delta_t^{h=H}$. The condition on $\mathcal{H}_t$ was required to come up with the mentioned decomposition through $\Delta_t^h$ and it is not needed anymore, therefore;

$$\mathbf{BayesReg}_T \leq \mathbb{E}\left[\sum_t^T \sum_h^H \phi(X_t^h, \pi_t(X_t^h))^\top (\widetilde{\omega}_t(\pi_t) - \omega^*)\right] \tag{12}$$

$$= \sum_t^T \mathbb{E}\left[\mathbb{1}^\top \mathbf{\Phi}_t (\widetilde{\omega}_t(\pi_t) - \omega^*)\right] \tag{13}$$

$$= \mathbb{E}\left[\sum_t^T \mathbb{1}^\top \mathbf{\Phi}_t \overline{\chi}_t^{-1/2} \overline{\chi}_t^{1/2} (\widetilde{\omega}_t(\pi_t) - \omega^*)\right] \tag{14}$$

$$\leq \mathbb{E}\left[\sqrt{H} \sum_t^T \|\mathbf{\Phi}_t\|_{\overline{\chi}_t^{-1}} \|\widetilde{\omega}_t(\pi_t) - \omega^*\|_{\overline{\chi}_t}\right] \tag{15}$$

$$\leq \mathbb{E}\left[\sqrt{H} \sum_t^T \|\mathbf{\Phi}_t\|_{\overline{\chi}_t^{-1}} 2\theta_{t-1}(\delta)\right] \tag{16}$$

Again similar to optimism we have the maximum expected cumulative reward condition on states of a episode is at most 1, we have;

$$\mathbf{BayesReg}_T \leq \sqrt{H}\mathbb{E}\left[\sum_t^T \min\{\|\mathbf{\Phi}_t\|_{\overline{\chi}_t^{-1}} 2\theta_{t-1}(\delta), 1\}\right]$$

$$\leq \sqrt{H}\mathbb{E}\left[\sum_t^T \sum_h^H 2\theta_{t-1}(\delta) \min\{\|\mathbf{\Phi}_t\|_{\overline{\chi}_t^{-1}}, 1\}\right]$$

Moreover, at time $T$, we can use Jensen's inequality, exploit the fact that $\theta_t(\delta)$ is an increasing function of $t$ and have

$$
\textbf{BayesReg}_T \leq 2\mathbb{E}\left[\sqrt{TH\theta_T(\delta)^2 \sum_t^T \min\{\|\mathbf{\Phi}_t\|_{\overline{\chi}_t^{-1}}, 1\}}\right]
$$

$$
\leq 2\left(\sigma\sqrt{2\log(1/\delta) + d\log(1 + TL^2/\lambda)} + \lambda^{1/2}L_\omega\right)\mathbb{E}\left[\sqrt{TH\sum_t^T \min\{\|\mathbf{\Phi}_t\|_{\overline{\chi}_t^{-1}}, 1\}}\right]
$$

$$
\leq 2\left(\sigma\sqrt{2\log(1/\delta) + d\log(1 + TL^2/\lambda)} + \lambda^{1/2}L_\omega\right)\sqrt{2THd\log(\lambda + TL^2/d)}
$$

$$(17)$$

Under event $\Theta_T$ which holds with probability at least $1 - \delta$. If we set $\delta = 1/T$ then the probability that the event $\Theta$ holds is $1 - 1/T$ and we get regret of at most the RHS of Eq. 17, otherwise with probability at most $1/T$ we get maximum regret of $T$, therefore

$$
\textbf{BayesReg}_T \leq 1 + 2\left(\sigma\sqrt{2\log(T) + d\log(1 + TL^2/\lambda)} + \lambda^{1/2}L_\omega\right)\sqrt{2THd\log(\lambda + TL^2/d)}
$$

and theorem statement follows.

### B.3 DISCOUNT FACTOR

Through the analysis of the theorems 1 and 2 we studied the regret upper bounds for undiscounted reward MDPs when $\gamma = 1$. If we consider the problem expression as follows

$$
\overline{\mathbf{R}}_t := \Psi\mathbf{R}_t \quad \overline{\nu}_t := \Psi\nu_t
$$

the parameter $\overline{\nu}_t$ for a general matrix $\Psi$ represent the stochasticity of the return under any discount factor $0 \leq \gamma \leq 1$. Under the general discount factor, the Assumption 1 describes the sub-Gaussian characteristics of $\overline{\nu}_t$. Moreover under discounted return, the value function represent the expectation of the discounted return, resulting in a potentially smaller $L_\omega$. Furthermore, in the decompositions derived in the Eqs. 9 and 12 regarding $\Delta_t^h$ we require to aggregate the regret while discounting the per step regrets. It means we replace $\mathbb{1}$ with $[1, \gamma, \gamma^2, \ldots, \gamma^{H-1}]^\top$ in those equations. Consequently, instead of considering the $\|\mathbb{1}\|_2$ to upper bound the regrets, we are interested in the discounted returns and consider $\|[1, \gamma, \gamma^2, \ldots, \gamma^{H-1}]\|_2$ to come up with the corresponding upper bounds and replace $\sqrt{H}$ in the final statements of the theorems 1 and 2 with a discount factor dependent and smaller value $\|[1, \gamma, \gamma^2, \ldots, \gamma^{H-1}]\|_2$.

### B.4 PROOF OF LEMMAS

**Lemma 3.** *Let $\alpha \in \mathcal{R}^d$ be be an arbitrary vector and for any $t \geq 0$ define*

$$
M_t^\alpha := \exp\left(\sum_i^t \left[\frac{\alpha^\top \mathbf{\Phi}_i^\top \overline{\nu}_i}{\sigma} - \frac{1}{2}\|\mathbf{\Phi_i}\alpha\|_2^2\right]\right)
$$

*Then, for a stopping time under the filtration $\{\mathcal{F}_t\}_{t=0}^\infty$, $M_\tau^\lambda \leq 1$.*

*Proof.* Lemma 3 We first show that $\{M_t^\alpha\}_{t=0}^\infty$ is a supermartingale sequence. Let

$$
D_i^\alpha = \exp\left(\frac{\alpha^\top \mathbf{\Phi}_i^\top \overline{\nu}_i}{\sigma} - \frac{1}{2}\|\mathbf{\Phi_i}\alpha\|_2^2\right)
$$

Therefore, we can rewrite $\mathbb{E}[M_t^\alpha]$ as follows:

$$
\mathbb{E}[M_t^\alpha|\mathcal{F}_{t-1}] = \mathbb{E}[D_1^\alpha \ldots D_{t-1}^\alpha D_t^\alpha|\mathcal{F}_{t-1}] = D_1^\alpha \ldots D_{t-1}^\alpha \mathbb{E}[D_t^\alpha|\mathcal{F}_{t-1}] \leq M_{t-1}^\alpha
$$

The last inequality follows since $\mathbb{E}[D_t^\alpha|\mathcal{F}_{t-1}] \leq 1$ due to Assumption 1, therefore since for the first time step $\mathbb{E}[M_1^\alpha] \leq 1$, then $\mathbb{E}[M_t^\alpha] \leq 1$. For a stopping time $\tau$, Define a variable $\overline{M}_t^\alpha = M_{\min\{t,\tau\}}^\alpha$ and since $\mathbb{E}[M_\tau^\alpha] = \mathbb{E}\left[\liminf_{t\to\infty}\overline{M}_t^\alpha\right] \leq \liminf_{t\to\infty}\mathbb{E}\left[\overline{M}_t^\alpha\right] \leq 1$, therefore the Lemma 3 holds. $\square$

**Lemma 4.** *[Extension to Self-normalized bound in Abbasi-Yadkori et al. (2011)] For a stopping time $\tau$ and filtration $\{\mathcal{F}_t\}_{t=0}^{\infty}$, with probability at least $1 - \delta$*

$$\|S_\tau\|^2_{\overline{\chi}_\tau^{-1}} \leq 2\sigma^2 \log(\frac{\det(\overline{\chi}_t)^{1/2} \det(\widetilde{\chi})^{-1/2}}{\delta})$$

*Proof.* of Lemma 4. Given the definition of the parameters of self-normalized process, we can rewrite $M_t^\alpha$ as follows;

$$M_t^\alpha = \exp\left(\frac{\alpha^\top S_t}{\sigma} - \frac{1}{2}\|\alpha\|_{\chi_t}\right)$$

Consider $\Omega$ as a Gaussian random vector and $f(\Omega = \alpha)$ denotes the density with covariance matrix of $\overline{\chi}^{-1}$. Define $M_t := \mathbb{E}\left[M_t^\Omega | \mathcal{F}_\infty\right]$. Therefore we have $\mathbb{E}[M_t] = \mathbb{E}\left[\mathbb{E}\left[M_t^\Omega | \Omega\right]\right] \leq 1$. Therefore

$$\begin{aligned}
M_t &= \int_{\mathbb{R}^d} \exp\left(\frac{\alpha^\top S_t}{\sigma} - \frac{1}{2}\|\alpha\|_{\chi_t}\right) f(\alpha) d\alpha \\
&= \int_{\mathbb{R}^d} \exp\left(\frac{1}{2}\|\alpha - \chi_t^{-1} S_t/\sigma\|^2_{\chi_t} + \frac{1}{2}\|S_t/\sigma\|^2_{\chi_t^{-1}}\right) f(\alpha) d\alpha \\
&= \sqrt{\frac{\det(\widetilde{\chi})}{(2\pi)^d}} \exp\left(\frac{1}{2}\|S_t/\sigma\|_{\chi_t^{-1}}\right) \int_{\mathbb{R}^d} \exp\left(\frac{1}{2}\|\alpha - \chi_t^{-1} S_t/\sigma\|^2_{\chi_t} + \frac{1}{2}\|\alpha\|^2_{\widetilde{\chi}}\right) d\alpha
\end{aligned}$$

Since $\chi_t$ and $\widetilde{\chi}$ are positive semi definite and positive definite respectively, we have

$$\begin{aligned}
\|\alpha - \chi_t^{-1} S_t/\sigma\|^2_{\chi_t} + \|\alpha\|^2_{\widetilde{\chi}} &= \|\alpha - \left(\widetilde{\chi} + \chi_t^{-1}\right) S_t/\sigma\|^2_{\widetilde{\chi}+\chi_t} + \|\chi_t^{-1} S_t/\sigma\|^2_{\chi_t} - \|S_t/\sigma\|^2_{(\widetilde{\chi}+\chi_t)^{-1}} \\
&= \|\alpha - \left(\widetilde{\chi} + \chi_t^{-1}\right) S_t/\sigma\|^2_{\widetilde{\chi}+\chi_t} + \|S_t/\sigma\|^2_{\chi_t^{-1}} - \|S_t/\sigma\|^2_{(\widetilde{\chi}+\chi_t)^{-1}}
\end{aligned}$$

Therefore,

$$\begin{aligned}
M_t &= \sqrt{\frac{\det(\widetilde{\chi})}{(2\pi)^d}} \exp\left(\frac{1}{2}\|S_t/\sigma\|_{(\widetilde{\chi}+\chi_t)^{-1}}\right) \int_{\mathbb{R}^d} \exp\left(\frac{1}{2}\|\alpha - \left(\widetilde{\chi} + \chi_t^{-1}\right) S_t/\sigma\|^2_{\widetilde{\chi}+\chi_t}\right) d\alpha \\
&= \left(\frac{\det(\widetilde{\chi})}{\det(\widetilde{\chi} + \chi_t)}\right)^{1/2} \exp\left(\frac{1}{2}\|S_t/\sigma\|^2_{(\widetilde{\chi}+\chi_t)^{-1}}\right)
\end{aligned}$$

Since $\mathbb{E}[M_\tau] \leq 1$ we have

$$\mathbb{P}\left(\|S_\tau\|^2_{(\widetilde{\chi}+\chi_\tau)^{-1}} \leq 2\sigma \log\left(\frac{\det(\widetilde{\chi} + \chi_\tau)^{1/2}}{\delta \det(\widetilde{\chi})^{1/2}}\right)\right) \leq \frac{\mathbb{E}\left[\exp\left(\frac{1}{2}\|S_\tau/\sigma\|^2_{(\widetilde{\chi}+\chi_\tau)^{-1}}\right)\right]}{\left(\frac{\det(\widetilde{\chi}+\chi_\tau)}{\delta \det(\widetilde{\chi})}\right)^{1/2}} \leq \delta$$

Where the Markov inequality is deployed for the final step. The stopping is considered to be the time step as the first time in the sequence when the concentration in the Lemma 4 does not hold. $\qquad \square$

*Proof of Lemma 1.* Given the estimator $\widehat{\omega}_t$ we have the following:

$$\begin{aligned}
\widehat{\omega}_t &= \left(\sum_i^t \mathbf{\Phi}_i^\top \mathbf{\Phi}_i + \lambda I\right)^{-1} \sum_i^t \left(\mathbf{\Phi}_i^\top \left(\mathbf{\Phi}_i \omega^* + \overline{\nu}_i\right)\right) \\
&= \left(\sum_i^t \mathbf{\Phi}_i^\top \mathbf{\Phi}_i + \lambda I\right)^{-1} \sum_i^t \left(\mathbf{\Phi}_i^\top \overline{\nu}_i\right) \\
&\quad + \left(\sum_i^t \mathbf{\Phi}_i^\top \mathbf{\Phi}_i + \lambda I\right)^{-1} \left(\sum_i^t \mathbf{\Phi}_i^\top \mathbf{\Phi}_i + \lambda I\right) \omega^* \\
&\quad - \lambda \left(\sum_i^t \mathbf{\Phi}_i^\top \mathbf{\Phi}_i + \lambda I\right)^{-1} \omega^*
\end{aligned}$$

therefore, for any vector $\zeta \in \mathbb{R}^d$

$$\zeta^\top \widehat{\omega}_t - \zeta^\top \omega^* = \zeta^\top \left( \sum_i^t \mathbf{\Phi}_i^\top \mathbf{\Phi}_i + \lambda I \right)^{-1} \sum_i^t \left( \mathbf{\Phi}_i^\top \overline{\nu}_i \right)$$
$$- \zeta^\top \lambda \left( \sum_i^t \mathbf{\Phi}_i^\top \mathbf{\Phi}_i + \lambda I \right)^{-1} \omega^*$$

As a results, applying Cauchy-Schwarz inequality and inequalities $\|\omega^*\|_{\overline{\chi}_t^{-1}} \leq \frac{1}{\lambda(\overline{\chi}_t)}\|\omega^*\|_2 \leq \frac{1}{\lambda}\|\omega^*\|_2$ we get

$$|\zeta^\top \widehat{\omega}_t - \zeta^\top \omega^*| \leq \|\zeta\|_{\overline{\chi}_t^{-1}} \| \sum_i^t \left( \mathbf{\Phi}_i^\top \overline{\nu}_i \right) \|_{\overline{\chi}_t^{-1}} + \lambda \|\zeta\|_{\overline{\chi}_t^{-1}} \|\omega^*\|_{\overline{\chi}_t^{-1}}$$

$$\leq \|\zeta\|_{\overline{\chi}_t^{-1}} \left( \| \sum_i^t \left( \mathbf{\Phi}_i^\top \overline{\nu}_i \right) \|_{\overline{\chi}_t^{-1}} + \lambda^{1/2}\|\omega^*\|_2 \right)$$

where applying self normalized Lemma 4, with probability at least $1 - \delta$

$$|\zeta^\top \widehat{\omega}_t - \zeta^\top \omega^*| \leq \|\zeta\|_{\overline{\chi}_t^{-1}} \left( 2\sigma \log \left( \frac{\det (\overline{\chi}_t)^{1/2}}{\delta \det (\widetilde{\chi})^{1/2}} \right) + \lambda^{1/2} L_\omega \right)$$

hold for any $\zeta$. By plugging in $\zeta = \overline{\chi}_t (\widehat{\omega}_t - \omega^*)$ we get the following;

$$\|\widehat{\omega}_t - \omega^*\|_{\overline{\chi}_t} \leq \theta_t(\delta) = \sigma \sqrt{2 \log \left( \frac{\det(\overline{\chi}_t)^{1/2} \det(\lambda I)^{-1/2}}{\delta} \right)} + \lambda^{1/2} L_\omega$$

The $\det(\overline{\chi}_t)$ can be written as $\det(\overline{\chi}_t) = \prod_j^d \alpha_j$ therefore, $trace(\overline{\chi}_t) = \sum_j^d \alpha_j$. We also know that;

$$(\prod_j^d \alpha_j)^{1/d} \leq \frac{\sum_j^d \alpha_j}{d}$$

A matrix extension to Lemma 10 and 11 in Abbasi-Yadkori et al. (2011) results in $\det(\overline{\chi}_t) \leq \left( \frac{trace(\overline{\chi}_t)}{d} \right)^d$ while we have $trace(\overline{\chi}_t) = trace(\lambda I) + trace(\sum_i^t \mathbf{\Phi}_i^\top \mathbf{\Phi}_i) \leq d\lambda + tL$,

$$\det(\overline{\chi}_t) \leq \left( \frac{d\lambda + tL}{d} \right)^d \tag{18}$$

therefore the main statement of Lemma 1 goes through. $\qquad \square$

*Proof.* Lemma 2 We have the following for the determinant of $\det (\overline{\chi}_T)$

$$\det (\overline{\chi}_T) = \det \left( \overline{\chi}_{T-1} + \mathbf{\Phi}_T^\top \mathbf{\Phi}_T \right)$$
$$= \det \left( I_H + \mathbf{\Phi}_T \overline{\chi}_t^{-1} \mathbf{\Phi}_T^\top \right) \det \left( \overline{\chi}_{T-1} \right)$$
$$= \prod_t^T \det \left( I_H + \mathbf{\Phi}_t \overline{\chi}_t^{-1} \mathbf{\Phi}_t^\top \right) \det \left( \widetilde{\chi} \right)$$

From linear algebra, and given a time step $t$, we define the characteristic polynomial of the matrix $\mathbf{\Phi}_t \overline{\chi}_t^{-1} \mathbf{\Phi}_t^\top$ as;

$$CP_t(\overline{\lambda}) = \det(\overline{\lambda} I_H - \mathbf{\Phi}_t \overline{\chi}_t^{-1} \mathbf{\Phi}_t^\top).$$

If the set $\{\lambda_1, \lambda_2, \ldots, \lambda_d\}$ represent the eigenvalues of $\mathbf{\Phi}_t \overline{\chi}_t^{-1} \mathbf{\Phi}_t^\top$ where $\lambda_i \geq \lambda_j$ if $i \geq j$, one can rewrite $CP_t(\overline{\lambda})$ as follows;

$$CP(\overline{\lambda}) = (\overline{\lambda} - \lambda_1)(\overline{\lambda} - \lambda_2) \cdots (\overline{\lambda} - \lambda_H)$$

Now by adding the matrix $I_H$ to $\mathbf{\Phi}_t \overline{\chi_t}^{-1} \mathbf{\Phi}_t^\top$, we get $CP_t'(\overline{\lambda}) = \det(\overline{\lambda} I_H - I_H - \mathbf{\Phi}_t \overline{\chi_t}^{-1} \mathbf{\Phi}_t^\top)$, therefore;

$$CP_t'(\overline{\lambda}) = (\overline{\lambda} - 1 - \lambda_1)(\overline{\lambda} - 1 - \lambda_2) \cdots (\overline{\lambda} - 1 - \lambda_H)$$

and the eigenvalues of $I_H + \mathbf{\Phi}_t \overline{\chi_t}^{-1} \mathbf{\Phi}_t^\top$ are shifted eigenvalues of $\mathbf{\Phi}_t \overline{\chi_t}^{-1} \mathbf{\Phi}_t^\top$ by $+1$, i.e. $\{\lambda_1 + 1, \lambda_2 + 1, \ldots, \lambda_d + 1\}$. Since $\xi^\top \mathbf{\Phi}_t \overline{\chi_t}^{-1} \mathbf{\Phi}_t^\top \xi = \left( \mathbf{\Phi} \overline{\chi_t}^{-1/2\top} \xi \right)^\top \left( \mathbf{\Phi} \overline{\chi_t}^{-1/2\top} \xi \right)$ all the eigenvalues $\{\lambda_1, \lambda_2, \ldots, \lambda_d\}$ are non-negative. As a results,

$$\det \left( I_H + \mathbf{\Phi}_t \overline{\chi_t}^{-1} \mathbf{\Phi}_t^\top \right) \geq \left( 1 + \|\mathbf{\Phi}_t\|_{\overline{\chi_t}^{-1}} \right)$$

As a consequence;

$$\det \left( \overline{\chi}_T \right) \geq \det \left( \widetilde{\chi} \right) \prod_t^T \left( 1 + \|\mathbf{\Phi}_t\|_{\overline{\chi_t}^{-1}} \right)$$

Moreover we have

$$2 \sum_t^T \log \left( 1 + \|\mathbf{\Phi}_t\|_{\overline{\chi_t}^{-1}} \right) \leq 2 \left( \log \left( \det \left( \overline{\chi}_T \right) \right) - \log \left( \det \left( \widetilde{\chi} \right) \right) \right) \leq 2d \log(\lambda + TL^2/d) \qquad (19)$$

and the statement follows.

$\square$

