# OpenReview forum: "Efficient Exploration through Bayesian Deep Q-Networks"
_ICLR.cc/2019/Conference_

### Official Review · AnonReviewer1 · 2018-10-24
**Lacks novelty, experiments incomplete, results misinterpreted. Clear reject.**

**Rating:** 2
**Confidence:** 5

**Review:**

The paper proposes performing Thompson Sampling (TS) using a Bayesian Linear Regressor (BLR) as the action-value function the inputs of which are parameterized as a deterministic neural net. The authors provide a regret bound for the BLR part of their method and provide a comparison against Double Deep Q-Learning (DDQL) on a series of computer games.

Strengths:
  * The paper presents some strong experimental results.

Weaknesses:
  * The novelty of the method falls a little short for a full-scale conference paper. After all, it is only a special case of [3] where the random weights are restricted to the top-most layer and the posterior is naturally calculated in closed form. Note that [3] also reports a proof-of-concept experiment on a Thompson Sampling setting.

  * Related to the point above, the paper should have definitely provided a comparison against [3]. It is hard to conclude much from the fact that the proposed method outperforms DDQN, which is by design not meant for sample efficiency and effective exploration. A DDQN with Dropout applied on multiple layers and Thompson Sampling followed as the policy would indeed be both a trivial design and a competitive baseline. Now the authors can argue what they provide on top of this design and how impactful it is.

  * If the main concern is sample efficiency, another highly relevant vein of research is model-based reinforcement learning. The paper should have provided a clear differentiation from the state of the art in this field as well.

  * Key citations to very closely related prior work are missing, for instance [1,2].

  * I have hard time to buy the disclaimers provided for Table 2. What is wrong with reporting results on the evaluation phase? Is that not what actually counts?

  * The appendix includes some material, such as critical experimental results, that are prerequisite for a reviewer to make a decision about its faith. To my take, this is not the Appendices are meant for. As the reviewers do not have to read the Appendices at all, all material required for a decision has to be in the body text. Therefore I deem all such essential material as invalid and make my decision without investigating them.

Minor:
  * The paper has excessively many typos and misspellings. This both gives negative signals about its level of maturity and calls for a detailed proofread.

[1] R. Dearden et al., Bayesian Q-learning, AAAI, 1998

[2] N. Tziortziotis et al., Linear Bayesian Reinforcement Learning, IJCAI, 2013

[3] Y. Gal, Z. Ghahramani, Dropout as a Bayesian Approximation: Representing Model Uncertainty in Deep Learning, ICML, 2016

---

> ### Author Response · Authors · 2018-11-14
> **The paper presents some strong experimental results.**
>
> We appreciate the thoughtful and detailed comments by the reviewer. In the following, we addressed the comments raised by the reviewer which helped us to revise the draft and make our paper more accessible.
>
>
> Regarding the Dropout: Drop-out, as another randomized exploration method is proposed by Gal & Ghahramani (2016), but Osband et al. (2016) argue about the deficiency of the estimated uncertainty and hardness in driving suitable exploration and exploitation trade-off from it. Please look at Appendix A in Osband et. al. 2016 https://arxiv.org/pdf/1602.04621.pdf.
> Osband el. Al. 2016 states that “The authors of Gal & Ghahramani (2016) propose a heteroskedastic variant which can help, but does not address the fundamental issue that for large networks trained to convergence all dropout samples may converge to every single datapoint... even the outliers.” This issue with dropout methods that they result in ensemble of many models but all almost the same is also observed in the adversarial attacks and defense community, e.g. Dhillon et. al. 2018 “ the dropout training procedure encourages all possible dropout masks to result in similar mappings.” Furthermore, after the reviewer’s comment, we also implemented DDQN-dropout and ran it on 4 randomly chosen Atari games (among those we ran for less than 50M time steps (please consider that these experiments are expensive)). We observed that the randomization in DDQN-dropout is deficient and results in a performance worse than DDQN on these 4 Atari games (the experiments are half way through and are still running, the statement is based on DDQN performance after seeing half of the data). We will add these further study in the final version of the paper.
>
> Regarding the model-based approaches: Model-based approaches are provably sample efficient, but they are mainly not scalable to the high dimensional settings.
>
> Regarding the mentioned papers: We appreciate your suggestions and added both of the mentioned papers to our paper.
>
>
> Regarding the Table2: As it is mentioned in the draft, the reported scores in table 2 are the scores directly reported in their original papers. As discussed in the paper, we are not aware of the detailed implementation and environment choices in this paper since their implementation codes are not publicly available. In order to see why the comparison through table 2 can be problematic, please, for example, look at the reported scores if DDQN in Bootstrapped DQN (Deep Exploration via Bootstrapped DQN ) and compare them with the reported score in the original DDQN paper. You can see that there is a huge gap. For example, some of them are as follows;
> Aline(2.9k,4k), Amidar(.7k,2.1k), Assault(5k,7k), Atlantis(65k,770k) where the first set of scores are DDQN scores in the original DDQN paper, and the second set of scores are the scores of DDQN, reported in the Bootstrap DQN paper. As you can see, direct reporting scores is not the best way of comparison and reasoning. Regarding the evaluation phase, we agree with the reviewer that scores in the evaluation phase are important when asymptotic performance is concerned but they are not sufficiently informative when regret and sample complexity are the measures of interest.
>
>
> We hope that the reviewer would kindly consider our replies, especially about the Dropout methods, and take them into account when assessing the final scores.

---

> > ### Author Response · Authors · 2018-11-18
> > **Dropout, as another randomized exploration method**
> >
> > Dear reviewer
> > We would like to bring your attention to the final results of the empirical study you kindly suggested. As we mentioned in our previous comment, we implemented the dropout version of the DDQN and compared its performance against BDQN, DDQN, DDQN+, as well as the random policy (the policy which chooses actions uniformly at random). We included the result of this empirical study to out paper.
> >
> > We would like to mention that we did not observe much gain beyond the performance of the random policy. Please see the discussion in the related works as well as in the section A.6). In the following we provided a summary of the empirical results on four randomly chosen games;
> >
> > Game		  BDQN	DDQN	DDQN+	 DropoutDDQN      RandomPolicy
> > CrazyClimber   124k       84k         102k               19k                            11k
> > Atlantis             3.24M     40k          65k                7.7k                           13k
> > Enduro              1.12k    0.38k        0.32k             0.27k                            0
> > Pong                   21         18.8           21                  -18                           -20.7
> >
> > As a conclusion, despite the arguments in the prior works on the deficieny of the dropout methods in providing reasonable randomizations in RL problems, we also empirically observed that dropout results in a performance worse than the performance of a plain epsilon-greedy DDQN and somewhat similar performance as the random policy's on at least four Atari games.

---

> ### Public Comment · ~Ian_Osband1 · 2018-11-28
> **I agree the paper should not be accepted, but this algorithm is not just a special case of "Dropout TS".**
>
> Following the discussion with the authors above, I would say that I don't think the characterization of their algorithm as "a special case of Dropout" is fair.
> Yes, both papers attempt to find an approximate form of posterior for Q-values... but I think this is one method (actually more similar to "RLSVI") and Dropout sampling on the final layer is another method.
> (Dearden et al, 1998) is another relevant comparison to add, but they do not correctly propagate uncertainty over multiple steps of TD error.
>
> I have personally had some concerns with the dropout-as-posterior, particularly for RL settings.
> Sections 2.1 and 2.2 of this paper https://arxiv.org/abs/1806.03335 outline some of these.
> Also, it looks like this paper provides more empirical evidence of DropoutTS poor performance.
>
> That said, I agree with many of your other points, and add severe concerns over the quality of the analysis

---

> > ### Comment · Area_Chair1 · 2018-11-28
> > **Agreed that this is not a special case of Dropout TS**
> >
> > I also disagreed with the reviewer's assessment here.  I don't agree with the reviewer's choice of relevant literature or demanded baselines.  This is why a fourth reviewer was brought in, and this review will be weighted accordingly.

---

### Official Review · AnonReviewer4 · 2018-10-30
**Appealing idea; poor delivery.**

**Rating:** 4
**Confidence:** 4

**Review:**

Summary: The paper proposes an approximate Thompson Sampling method for value function learning when using deep function approximation.

Research Context: Thompson Sampling is an algorithm for sequential decision making under uncertainty that could provide efficient exploration (or lead to near optimal cumulative regret) under some assumptions. The most critical one is the ability to sample from the posterior distribution over problem models given the already collected data. In most cases, this is not feasible, so we need to rely on approximate posteriors, or, informally, on distributions that somehow assign reasonable mass to plausible models. The paper tries to address this.

Main Idea: In particular, the idea here is to simultaneously train a deep Q network while choosing actions based on samples for the linear parameters of the last layer. On one hand, this seems sensible: a distribution over the last layer weights provides an approximate posterior over Q functions, as needed, and a linear model could work after learning an appropriate representation. On the other, this seems doable: there are close-form updates for Bayesian linear regression when using a Gaussian prior and likelihood, as proposed in the paper.

Pros:
- Simple and elegant algorithm.
- Strong empirical results in standard benchmarks.

Cons:
- The paper is very poorly written; the number of typos is countless, and in general the paper is quite hard to read and to follow.
- I share the concerns expressed in the first public comment regarding the correctness of the theoretical statements (Theorem 1) or, at least, the proposed proofs. Notation is very hard to parse, and the meaning of some claims is not clear ('the PSRL on w', 'we use w instead of w', 'the estimated \hat{b_t}', '\pi_t(x, a) = a = ...'). I'd appreciate a clear proof strategy outline. In addition, it'd be quite useful if the authors could highlight the specific technical contributions of the proposed analysis, and how they rely on and relate to previous analyses (Abbasi-Yadkori et al., De la Peña et al., Osband et al., etc).
- I think Table 1, Figure 1, and Figure 2 are not particularly useful and could be removed.

Questions:
- Last year, there was a paper published in ICLR [1] that proposed basically the same algorithm for contextual bandits. They reported as essential to also learn the noise levels for different actions, while in this work \sigma_\epsilon is assumed known, fixed, and common across actions (see paragraph to the left of Figure 2). I'm curious why not learn it for each action using an Inverse-Gamma prior as proposed in [1], or if this was actually something you tried, and what the performance consequences were. In principle, my hunch is it should have a strong impact on the amount of exploration imposed by the algorithm (see Equation 3) over time.
- A minor comment: the dimension 'd' in Theorem 1 is a *design choice* in the proposed algorithm. Of course, Theorem 1 relies on some assumptions that may be harder to satisfy for decreasing values of 'd', but I think some further comment can be useful as some readers may think the theorem is indeed suggesting we should set as small 'd' as possible...
- More generally, what are the expected practical consequences of the mismatch between the proposed algorithm (representation is learned alongside with linear TS) and the setup in Theorem 1 (representation is fixed or known, and prior and likelihood are not misspecified)?

Conclusion:
While definitely a promising direction, the paper requires significant further work, writing improvement, and polishing. At this point, I'm unable to certify that the theoretical contribution is correct.


I'm willing to change my score if some of the comments above are properly addressed. Thanks.





[1] - Deep Bayesian Bandits Showdown: An Empirical Comparison of Bayesian Deep Networks for Thompson Sampling.

---

> ### Author Response · Authors · 2018-11-14
> **Simple and elegant algorithm +  Strong empirical results in standard benchmarks.**
>
> We would like to thank the reviewer for the helpful comments. We appreciate the comments and believe that they significantly improve the clarity of our paper.
>
>
> At first, we would like to apologize for the typos. We addressed them based on the extensive review by AnonRev2.
>
> Regarding the proof: Based on Ian Osband’s and AnonRev2 comments, we polished the proof and believe it is now in a good shape. We would like to bring the reviewer’s attention to the revised version of the Appendix where we further improved the clarity of the expressions and derivations.
>
>
> Regarding the noise model: As the reviewer mentioned, the approach in the mentioned proceeding paper [1] is similar to BDQN (BDQN was publicly available as a workshop paper before [1] appearance). As the reviewer also mentioned, it is an interesting approach to estimate the noise level which could be helpful in practice. We would like to bring the reviewer’s attention to our Lemma 1 and Eq3 where we show that the noise level is just a constant scale of the confidence interval which vanishes as O(1/sqrt(t)). Therefore, the noise level does not have a critical and deriving effect when the confidence interval is small. To be more accurate, it is also worth noting that one can deploy related Bernstein-based approaches as in “Minimax Regret Bounds for Reinforcement Learning” for the noise estimation, but this approach results in the more complicated algorithm as well as an additional fixed but intolerably big term (cubic in the dimension) in the regret.
>
>
> Regarding the design choice “d”: We apologize for the lack of clear clarification that “d” is a design choice in the theorem and cannot be set to a small value unless the assumption holds. We restated this statement in the revised draft and made it clear.
>
>
> Regarding the connection between the linear TS and theorem 1: If the prior and posterior are both Gaussian, similar to the linear Bandits case studied in Russo2014 “Learning to optimize via posterior sampling”, the linear TS is equivalent to the PSRL algorithm mentioned in Theorem 1.
>
>
> We appreciate the reviewer’s comments and believe they help to improve the current state of the paper. We would be grateful if the reviewer could look at the new uploaded draft.

---

### Official Review · AnonReviewer2 · 2018-11-01
**Major clarity issues**

**Rating:** 4
**Confidence:** 2

**Review:**

Update after feedback: I would like to thank the authors for huge work done on improving the paper. I appreciate the tight time constrains given during the discussion phase and big steps towards more clear paper, but at the current stage I keep my opinion that the paper is not ready for publication. Also variability of concerns raised by other reviewers does not motivate acceptance.

I would like to encourage the authors to make careful revision and I would be happy to see this work published. It looks very promising.

Just an example of still unclear parts of the paper: the text between eq. (3) and (4). This describes the proposed method, together with theoretical discussions this is the main part of the paper. As a reader I would appreciate this part being written detailed, step by step.
=========================================================

The paper proposes the Bayesian version of DQN (by replacing the last layer with Bayesian linear regression) for efficient exploration.

The paper looks very promising because of a relatively simple methodology (in the positive sense) and impressive results, but I find the paper having big issues with clarity. There are so many mistakes, typos, unclear statements and a questionable structure in the text that it is difficult to understand many parts. In the current version the paper is not ready for publication.

In details (more in the order of appearance rather than of importance):
1. It seems that the authors use “sample” for tuples from the experience replay buffer and draws W from its posterior distribution (at least for these two purposes), which is extremely confusing
2. pp.1-2 “We show that the Bayesian regret is bounded by O(d \sqrt{N}), after N time steps for a d-dimensional feature map, and this bound is shown to be tight up-to logarithmic factors.” – maybe too many details for an abstract and introduction and it is unclear for a reader anyway at that point
3. p.1 “A central challenge in reinforcement learning (RL) is to design efficient exploration-exploitation tradeoff” – sounds too strong. Isn’t the central challenge to train an agent to get a maximum reward? It’s better to change to at least “One of central challenges”
4. p.1 “ε-greedy which uniformly explores over all the non-greedy strategies with 1 − ε probability” – it is possible, but isn’t it more conventional for an epsilon-greedy policy to take a random action with the probability epsilon and acts greedy with the probability 1 – epsilon? Moreover, later in Section 2 the authors state the opposite “where with ε probability it chooses a random action and with 1 − ε probability it chooses the greedy action based on the estimated Q function.”
5. p.1 “An action is chosen from the posterior distribution of the belief” – a posterior distribution is the belief
6. p.2 “and follow the same target objective” – if BDQN is truly Bayesian it should find a posterior distribution over weights, whereas in DDQN there is no such concept as a posterior distribution over weights, therefore, this statement does not sound right
7. p.2 “This can be considered as a surrogate for sample complexity and regret. Indeed, no single measure of performance provides a complete picture of an algorithm, and we present detailed experiments in Section 4” – maybe too many details for introduction (plus missing full stop at the end)
8. p.2 “This is the cost of inverting a 512 × 512 matrix every 100,000 time steps, which is negligible.” – doesn’t this depend on some parameter choices? Now the claim looks like it is true unconditionally. Also too many details for introduction
9. p.2 “On the other hand, more sophisticated Bayesian RL techniques are significantly more expensive and have not lead to large gains over DQN and DDQN.” – it would be better to justify the claim with some reference
10. Previous work presented in Introduction is a bit confusing. If the authors want to focus only on Thompson Sampling approaches, then it is unclear, why they mentioned OFU methods. If they mention OFU methods, then it is unclear why other exploration methods are not covered (in Introduction). It is better to either move OFU methods to Related Work completely, or to give a taste of other methods (for example, from Related Work) in Introduction as well
11. p.3 “Consider an MDP M as a tuple <X , A, P, P0, R, γ>, with state space X , action space A, the transition kernel P, accompanied with reward function of R, and discount factor 0 ≤ γ < 1.” – P_0 is not defined
12. p.4 “A common assumption in DNN is that the feature representation is suitable for linear classification or regression (same assumption in DDQN), therefore, building a linear model on the features is a suitable choice.” – the statement is more confusing than explaining. Maybe it is better to state that the last fully connected layer, representing linear relationship, in DQN is replaced with BLR in the proposed model
13. p.5 In eq. (3) it is better to carry definition of $\bar{w}_a$ outside the Gaussian distribution, as it is done for $\Xi_a$
14. p.5 The text between eq. (3) and (4) seems to be important for the model description and yet it is very unclear: how $a_{TS}$ is used? “we draw $w_a$ follow $a_{TS}$” – do the authors mean “following” (though it is still unclear with “following”)? What does notation $[W^T \phi^{\theta} (x_{\tau})]_{a_{\tau}}$ denote? Which time steps do the authors mean?
15. p.5 The paragraph under eq. (4) is also very confusing. “to the mean of the posterior A.6.” – reference to the appendix without proper verbal reference. Cov in Algorithm 1 is undefined, is it equal to $\Xi$? Notation in step 8 in Algorithm 1 is too complicated.
16. Algorithm 1 gives a vague idea about the proposed algorithm, but the text should be revised, the current version is very unclear and confusing
17. pp.5-6 The text of the authors' attempts to reproduce the results of others' work (from "We also aimed to implement..." to "during the course of learning and exploration") should be formalised
18. p. 6 "We report the number of samples" - which samples? W? from the buffer replay?
19. p. 6 missing reference for DDQN+
20. p. 6 definition of SC+ and references for baselines should be moved from the table caption to the main text of the paper
21. p. 6 Table 3 is never discussed, appears in a random place of the text, there should be note in its reference that it is in the appendix
22. p.6 Where is the text for footnotes 3-6?
23. p.6 Table 2 may be transposed to fit the borders
24. p.6 (and later) It is unclear why exploration in BDQN is called targeted
25. p.7 Caption of Figure 3 is not very good
26. p.7 Too small font size of axis labels and titles in plots in Figure 3 (there is still a room for 1.5 pages, moreover the paper is allowed to go beyond 10 pages due to big figures)
27. p.7 Figure 3. Why Assault has different from the others y-axis? Why in y-axis (for the others) is "per episode" and x-axis is "number of steps" (wise versa for Assault)?
27. Section 5 should go before Experiments
28. p. 7 “Where Ψ is upper triangular matrix all ones 6.” – reference 6 should be surrounded by brackets and/or preceded by "eq." and it is unclear what “all ones” means especially given than the matrix in eq. (6) does not contain only ones
29. p. 7 “Similar to the linear bandit problems,” – missing citation
30. p. 7 PSRL appears in the theorem, but is introduced only later in Related work
31. p. 7 “Proof: in Theorem. B” – proof is given in Appendix B?
32. p. 8 Theorem discussion, “grows not faster than linear in the dimension, and \sqrt(HT)” – unclear. Is it linear in the product of dimension (of what?) and \sqrt(HT)?
33. p.8 “On lower bound; since for H = 1…” – what on lower bound?
34. p.8 “our bound is order optimal in d and T” – what do the authors mean by this?
35. p.8 "while also the state of the art performance bounds are preserved" - what does it mean?
36. p.8 "To combat these shortcomings, " - which ones?
37. p.8 "one is common with our set of 15 games which BDQN outperformS it..." - what is it?
38. p.9 "Due to the computational limitations..." - it is better to remove this sentence
39. p.9 missing connection in "where the feature representation is fixed, BDQN is given the feature representation", or some parts of this sentence should be removed?
40. p.9 PAC is not introduced
41. pp.13-14 There is no need to divide Appendices A.2 and A.3. In fact, it is more confusing than helpful with the last paragraph in A.2 repeating, sometimes verbatim, the beginning of the first paragraph in A.3
42. In the experiments, do the authors pre-train their BDQN with DQN? In this case, it is unfair to say that BDQN learns faster than DDQN if the latter is not pre-trained with DQN as well. Or is pre-training with DQN is used only for hyperparameter tuning?
43. p.14 “Fig. 4 shows that the DDQN with higher learning rates learns as good as BDQN at the very beginning but it can not maintain the rate of improvement and degrade even worse than the original DDQN.” – it seems that the authors tried two learning rates for DDQN, for the one it is clear that it is set to 0.0025, another one is unclear. The word “original” is also unclear in this context. From the legend of Figure 4 it seems that the second choice for the learning rate is 0.00025, but it should be stated in the text more explicitly. The legend label “DDQN-10xlr” is not the best choice either. It is better to specify explicitly the value of the learning rate for both DDQN
44. p.15 “As it is mentioned in Alg. 1, to update the posterior distribution, BDQN draws B samples from the replay buffer and needs to compute the feature vector of them.” – B samples never mentioned in Algorithm 1
45. p.15 “during the duration of 100k decision making steps, for the learning procedure,” – i) “during … duration”, ii) what did the authors meant by “decision making steps” and “the learning procedure”?, and iii) too many commas
46. p.15 “where $\tilde{T}^{sample}$, the period that of $\tilde{W}$ is sampled our of posterior” – this text does not make sense. Is “our” supposed to be “out”? “… the number of steps, after which a new $\tilde{W}$ is sampled from the posterior”?
47. p.15 “$\tilde{W}$ is being used just for making Thompson sampling actions” – could the authors be more specific about the actions here?
48. p.16 “In BDQN, as mentioned in Eq. 3, the prior and likelihood are conjugate of each others.” – it is difficult to imagine that an equation would mention anything and eq. (3) gives just the final formula for the posterior, rather than the prior and likelihood
49. p.16 The formula after “we have a closed form posterior distribution of the discounted return, ” is unclear
50. p.17 “we use ω instead of ω to avoid any possible confusion” – are there any differences between two omegas?
51. p.17 what is $\hat{b}_t$?

There are a lot of minor mistakes and typos also, I will add them as a comment since there is a limit of characters for the review.

---

### Official Review · AnonReviewer3 · 2018-11-04
**-**

**Rating:** 6
**Confidence:** 2

**Review:**

This paper proposes a method for more efficient exploration in RL by maintaining uncertainty estimates over the learned Q-value function. It is comparable to Double DQN (DDQN) but uses its learned uncertainty estimates with Thompson sampling for exploration, rather than \epsilon-greedy. Empirically the method is significantly more sample-efficient for Atari agents than DDQN and other baselines.

=====================================

Pros:

Introduction and preliminaries section give useful background context and motivation. I found it easy to follow despite not having much hands-on background in RL.

Proposes a novel (to my knowledge) exploration method for RL which intuitively seems like it should work better than \epsilon-greedy exploration. The method looks simple to implement on top of existing Q-learning based methods and has minimal computational and memory costs.

Strong empirical performance as compared with appropriate baselines -- especially to DDQN(+) where the comparison is direct with the methods only differing in exploration strategy.

Good discussion of practical implementation issues (architecture, hyperparameters, etc.) in Appendix A.

=====================================

Cons/questions/suggestions/nitpicks:

Algorithm 1 line 11: “Update W^{target} and Cov” -- how? I see only a brief mention of how W^{target} is updated in the last paragraph of Sec. 3, but it’s not obvious to me how the algorithm is actually implemented from this, and I don’t see any mention of how Cov is updated.

Algorithm 1: I’d like to know more about how sample-efficiency varies with T^{sample} given that T^{sample}>1 is doing something other than true Thompson sampling. Does the regret bound hold with T^{sample}>1? Also, based on the discussion in Appendix A, approximating the episode length seems to be the goal in choosing a setting of T^{sample} -- so why not just always resample at the beginning of each episode instead of using a constant T^{sample}?

Theorem 1: I don’t understand the Theorem statement (let alone the proof) or what it tells me about the proposed BDQN algorithm. First, as a “non-RL person” I wasn’t familiar with PSRL, but I see it’s later defined as “posterior-sampling RL”. This should be clarified earlier for readers that aren't familiar with this line of work. But this still doesn’t fully explain what “the PSRL on \omega” means. If it means you follow an existing PSRL algorithm to learn \omega, then how does the theorem relate to the proposed algorithm? I'm sure I'm missing something but the connection was unclear to me.

Theorem 1: there’s a common abuse of big-O notation here that should be fixed for a formal statement -- O(f(n)) by definition is a set corresponding to an upper-bound, so this should probably be written as g(n) \in O(f(n)) rather than g(n)<=O(f(n)). (Or alternatively, just rewritten without big-O notation.)

Table 2: should be reformatted to make it clear that the rightmost 3 columns are not additional baseline methods (e.g. adding a vertical line would be good enough).

Appendix A.8, “A discussion on safety”: this section should either be much more fleshed out or removed. I didn’t understand the statement at the end at all -- “one can... come up with a criterion for safe RL just by looking at high and low probability events” -- huh? What is even meant by “safe RL” in this context? Nothing is referenced.

Overall, much of the writing seems quite rushed with many typos and grammatical errors throughout. This should be cleaned up for a final version. To give a particularly common example, there are many inline references that do not fit in the sentence and distract from the flow -- these should be changed to \citep.

How does this compare with “A Distributional Perspective on Reinforcement Learning” (Bellemare et al., ICML 2017) both in terms of the approach and performance? The proposed method seems to at least superficially share motivation with this work (and uses the same Atari benchmark, as far as I can tell) but it is not discussed or compared.

=====================================

Overall, though many parts of the paper could use significant cleanup and clarification, the paper proposes a novel yet relatively simple and intuitive approach with strong empirical performance gains over comparable baselines.

---

> ### Author Response · Authors · 2018-11-14
> **Strong empirical performance as compared with appropriate baselines - More clear description for the main Algorithm.**
>
>
> We would like to thank the reviewer for taking the time to leave a thoughtful review. In the following, we addressed the comments raised by the reviewer which helped us to revise the draft and make it more accessible.
>
> Regarding the posterior updates: We apologize for the lack of clarity. The Cov is computed by applying Bayesian Linear regression on a subset of randomly sampled tuples in the experience replay buffer. We expressed it in a more detail in Eq.3 of the new draft. Regarding the updates of target models: Similiar to DQN paper, we update the target feature network every T^{target} and set its parameters to the feature network parameter. We also update the linear part of the target model, i.e., w^{target}, every  T^{Bayes target} and set the weights to the mean of the posterior distribution.
>
> Regarding the Thompson Sampling in Algorithm 1: As the reviewer also pointed out, Thompson sampling in multi-arm bandits can be more efficient if we sample from the posterior distribution at each time step, i.e., T^{sample}=1. But as it has been proven in our Theorem1 as well as Osband et al. 13, and Osband et al. 14, the true Thompson sampling can have T^{sample}>1. Moreover, as it is expressed in Appendix A.5, as long as T^{sample} is in the same order of the horizon length, the choice of T^{sample} does not critically affect the performance. As the reviewer also mentioned, sampling from the posterior at the beginning of each episode could marginally enhance the performance. While sampling at the beginning of episodes does not dramatically affect the BDQN performance, it provides additional information to the algorithm about the games settings which might cause a controversy in the fair comparison against other algorithms.
>
> We apologize for the lack of a concrete definition of PSRL, and we agree with the reviewer that it should have been clearly defined. We added a clear definition of PSRL as well as a new algorithm block (now Alg2). The theoretical contribution in this paper suggests that if an RL agent follows the posterior sampling over weights w for exploration and exploitation trade-off, the agent’s regret is upper bounded by \tilde{O(d\sqrt(T))}. Similar to the approach in Russo et al. 14 for linear bandits, if the prior and the likelihood are conjugates of each other and Gaussian, then BDQN is equivalent to PSRL.
> “A side point: It is an interesting observation that in the proof of our theorems, we construct self-normalized processes which result in a Gaussian approximation of confidence. The Gaussian approximation of confidence also has been deployed for linear bandits in “Linear Thompson Sampling Revisited”. Therefore, the choice of Gaussian is well motivated.”
>
> We appreciate the comment by the reviewer on the O notation. We fixed it in the new draft.
>
> Regarding the Table 2: we added the vertical lines to both tables, the table 2 and the table 3.
>
> Regarding the safety discussion in A.8: We apologize if the statement was not clear in the discussion on safety. We added a detailed explanation in addition to a new figure for the proof-of-concept to clarify our statement. Generally, in the study of safety in RL, a RL agent avoids taking actions with high probability of low return. If the chance of receiving a low return under a specific action is high, then that action is not a favorable action.  In appendix A.8 we show how the approaches studied in this paper also approximate the distribution over the return which can further be used for the safe exploration.
>
> Typos: Thanks for pointing it out. Also thanks to the reviewer 2, we addressed the typos in the new draft.
>
>
> Regarding the Distributional RL: The approach in Bellmare et al. 2017 approximates the distribution of the return (\sum_t \gamma r_t|x,a) rather the distribution (or uncertainty) over the expected return Q(x,a)=E[\sum_t \gamma r_t|x,a]. It is worth reminding that the mean of the return distribution is the Q function. Conceptually, approximating the distribution of the return is a redundant effort if our goal is to approximate the Q function. The approach in Bellmare et al. 2017 proposes first to deploy a deep neural network and approximate the return distribution, then apply a simple bin-based discretization technique to compute the mean of the approximated distribution, which is again the approximated Q function. Interestingly, Bellmare et al. 2017 empirically show that this approach results in a better approximation of the Q function. This approach is a variant of the Q-learning algorithm.

---

### Author Response · Authors · 2018-10-19
**A post-submission typo in the second paragraph of the introduction**

Dear reviewers and readers

We, unfortunately, noticed we made a typo post-submission in the second paragraph of the introduction section. In particular, this typo has appeared in the starting sentence,
"An alternative to optimism-under-uncertainty is Thompson Sampling (TS), a general sampling and randomizathttps://www.overleaf.com/1332425641pdxdghynmdhyion approach (in both frequentist and Bayesian settings) (Thompson, 1933)."
which should be replaced with
"An alternative to optimism-under-uncertainty is Thompson Sampling (TS), a general sampling and randomization approach (in both frequentist and Bayesian settings) (Thompson, 1933)."

We have already addressed this issue in our draft and apologize for any inconvenience this causes.

Sincerely yours
Authors

---

### Public Comment · ~Ian_Osband1 · 2018-10-23
**Significant concerns with the analytical regret bound**

## Overview

This paper appears to be a revised and updated submission from ICLR 2018 https://openreview.net/forum?id=Bk6qQGWRb&noteId=rJMiU16HM.
At a high level, this paper offers a particular adaptation of RLSVI to deep neural network models where a linear approximation is used on only the final layer.

Perhaps the key addition to this paper comes from the novel theoretical results and, in particular, the regret bound of Theorem 1.
This result would be *highly* significant if correct, as some of the first polynomial regret bounds for value-based RL with function approximation.
We should note it would not be the first Bayesian regret bounds beyond tabula rasa, even for the PSRL algorithm (e.g. https://arxiv.org/abs/1406.1853, https://arxiv.org/abs/1403.3741, https://arxiv.org/abs/1709.04047)... but this doesn't mean it would not be *very impressive* result.

However, upon close examination I am highly skeptical of Theorem 1 (at least in its current presentation).
At the very least, the statement/proof as presented in this paper is insufficiently clearly-stated for verification.
In fact, I actually believe there are several important missing pieces to this analysis that mean this approach is not correct...


## Issues with "PSRL" analysis

Critiquing specific parts of the analysis is difficult, since the analysis is not broken into separable pieces.
I would encourage the authors to refactor the analysis into a more step-by-step argument, so that issues can be isolated more carefully... even if this results in a long appendix.

Specific comments:

(a) - The theorem statement conflates PSRL (that samples from the exact posterior distribution) with RLSVI (that uses a Gaussian approximation to the TD error) with "Bayesian DQN" (which appears to be RLSVI when used with linear approximation). It's quite hard to know exactly what algorithm is meant by "The PSRL on w", since this is not defined.
If this algorithm refers to RLSVI we should note that samples from RLSVI are *not* drawn from the posterior, but instead use a Gaussian approximation to the Bellman residual. Specifically this argument is used at the bottom of p19 to set a term = 0 in expectation, but will not generally be true. Previous analyses of RLSVI make special care to address this mismatch but only in a tabular setting (e.g. https://arxiv.org/abs/1703.07608).

(b) - In the analysis the coefficient vector "w" parameterizes not only optimal Q, but also the specific MDP model (reward R, transition T). However, parameterizing by w means the underlying model are not well-defined.

(c) - BayesRegret takes expectation over all variables, therefore there should be no delta in Theorem 1.

(d) - The Theorem statement makes no mention of discounting, but the analysis given in terms of gamma, which feels wrong.


Looking through some of the appendix there are several other issues (e.g. Lemma 2 misses a square in the first term of the proof), but digging into these particular issues too deeply probably isn't very helpful at this stage.
It seems like this was maybe a little rushed generally ("we use w instead of w to avoid any possible confusion" p.17).
I would suggest a clean rewrite of the Theorem statement + proof so that anyone can easily verify each step.

## Summary

My belief is that there are some significant missing pieces to this analysis and it will not be possible to remedy them without significant additional tools/insight.

---

> ### Author Response · Authors · 2018-10-27
> **Clarifications on the regret analysis.**
>
> Dear Ian
> Thank you for your interest in this work, and I appreciate your comments. They were helpful to improve the representation in the appendix.
>
> Regarding the structure of analysis, I followed the flow in Abbasi-Yadkori et al 2010 to make the theoretical contribution more transparent. Upon your feedback, I changed the presentation in the appendix and refactored pieces such that the appendix is more accessible. The Lemmas and their proofs are factored out from the main body of the theorem proof. More explanation in the components and more detailed comments on derivation are also provided.
>
> a) PSRL-RLSVI: I believe the main confusion is due to the fact that I was not clear enough in stating that the theoretical analysis is for PSRL where the agent knows the exact prior and likelihood, and I apologize for the confusion. If the prior and likelihood are Gaussian, then BDQN (for the fixed representation up to some modification) is equivalent to PSRL; otherwise, BDQN approximates the posterior distribution with a Gaussian distribution, and the bound does not hold. I also added the PSRL algorithm block into the main text. In short, at the beginning of each episode, the algorithm draws a sample w_t from the w posterior distribution and follows w_t’s policy. The prior and posterior need not to be Gaussian, rather known.
>
> b) Mapping from MDP to w: We consider a class of MDPs for which the optimal Qs are linear. By definition, given an underlying MDP, there exists a w^* (up to some conditions). Clearly, as you also mentioned, the mapping from MDP to w, in general, cannot be defined as a bijective mapping. Therefore, I would avoid saying w specifies the model and instead the other way, as it is also mentioned in the paper. In order to prevent any further misunderstanding, I explained it in more detail and also changed a few lines to clarify it the most.
> Moreover, in order to prove the theorem, there is no need to bring the model in the proof picture. I explained it through the model to ease the understanding of the derivation. I admit that I could explain the derivation in a better and clearer way. You can see that the same derivation can be done by adding and subtraction through Eq 6, the linear model. In order to prevent any confusion and directly carry the message, I wrote the derivation without bringing the MDP model into the proof picture in the new draft. Thanks for this comment, the current derivation without MDP model is now more transparent.
>
> c) High probability bound: When we use frequentist argument for a bound, we usually get high probability bound for either Bayesian or frequentist, e.g. “McAllester 1998
> Some PAC-Bayesian Theorems”
> As you know, when one substitutes \delta with 1/T (or sometimes 1/T^2) we get log(T) instead of log(1/\delta) in the bound as well as additional positive constant  T/T Loss_max in the final bound. For example, your paper Osband et al 2013 and Russo et al 2014 follow the same argument. But the bound is not “any time”anymore. In order to simplify the theorem, I set \delta = 1/T to match the claim in Osband et al 2013.
>
> d) [General discount factor set] I apologize for the confusion. In the main text I talk about a discount factor of 1 (undiscounted), but in the appendix, I define the discount factor \gamma to be in a closed set of [0, 1]. Please note that the upper bound on the discount factor is 1 and 1 is in the set, i.e., it contains \gamma=1. So it should feel more general. For simplicity, I first derived the regret bound for \gamma=1 then showed it is extendable to any \gamma. I elaborated more in the new draft on how to extend the current analyses to any 0\geq\gamma\geq 1.
>
>
> Thank you for pointing out the typo in the in Lemma 2, I fixed that.
>
>
> The new draft is re-organized and is much more accessible. I’ll upload it to openreview when the website is open again. It would be great if you could look at it and point out the part which requires more clarification in the new draft. It would be again helpful to have your feedback on it. They are helping to improve the accessibility of the proof.
>
> Sincerely yours
> Authors

---

> > ### Author Response · Authors · 2018-11-14
> > **Updated draft-factored Appendix**
> >
> > Dear Ian,
> > I would like to inform you that I uploaded the revised version of the draft as promised. Based on your and the reviewers' great comments, I significantly improved the Appendix and with high probability, I think it is now much more clear and factored. I would be grateful to have your insightful feedbacks again.

---

### Author Response · Authors · 2018-11-14
**General reply to reviewers**

Dear reviewers
We would like to thank the reviewers for taking the time to leave thoughtful reviews. Given these feedbacks, we have significantly improved the draft and hope the reviewers will take this into account when assessing the final scores. We appreciate the reviewers for the time and effort they dedicated to our paper. Please find individual replies to each of the reviews in the respective threads. Based on the reviewers' reviews and the comment by Ian Osband we revised the draft and uploaded the new version.

---

> ### Public Comment · ~Ian_Osband1 · 2018-11-21
> **Fundamental issues with proof are still not solved**
>
> Thank you to the authors for trying to improve the paper and analysis.
> Some parts of the paper have improved, but there are still many parts that are difficult to follow.
> (e.g. confidence set C_t is not introduced before Appendix, gamma discount appears in undiscounted analysis)
>
> Rather than get bogged down in small details I want to highlight at least one fundamental error in the analysis of PSRL (Theorem 1).
> Stating this clearly should be enough to convince a third party that this analysis needs more work.
>
>
> ## Main theorem claim + prior/posterior disconnect
>
> The authors claim "the first model-free theoretical guarantee for continuous state-action space MDP, beyond the tabular setting".
> This means that, the PSRL algorithm of Theorem 1, should maintain its posterior over $w$ without an underlying model.
>
> The description of the PSRL algorithm (and associated regret bound) is not tied to any specific choice of prior/likelihood format.
> This statement is itself unclear, do the authors mean this result to hold for all choices of prior/likelihood, or specifically using a Gaussian form for PSRL updates?
> Either way, the application of the "posterior sampling lemma" on page 23 (that conditioned on any data H_t, the sampled  posterior is identically distributed to the optimal value) is inconsistent.
>
> There are two main options here:
>
>   a - If their "PSRL" is using a model-free Gaussian form of the optimal value (per Gaussian linear bandits), then this is not the correct posterior for the Bayesian decision rule on the optimal policy for all underlying MDPs.
>   To see this, note that the *optimal* policy includes a max operator over actions, this breaks Gaussian conjugacy even if the rewards are gaussian... this is the main difficulty in prior analyses of RLSVI (e.g. https://arxiv.org/abs/1402.0635).
>
>   b - If PSRL is using the correct form of the posterior, then it must be using the information about the likelihood of the "noise process" $\nu$ and thus this is not a model-free algorithm.
>   This setting is most similar to prior work on PSRL with generalization (e.g. https://arxiv.org/abs/1406.1853, https://arxiv.org/abs/1709.04047)
>
> To remedy this, the authors need to be much more clear about what prior/posterior sampling procedure their PSRL algorithm uses *and* the prior/likelihood of tasks against which they are assessing their algorithm.
> If the two are the same, then they need to be clear on *how* PSRL is able to be model-free algorithm and still match the exact posterior of the underlying MDP given any possible data H_t.
>
> This is an interesting line of research, and it would be impactful to get this answer right!
> Unfortunately, I do not think this proof is correct and so it should not be accepted.

---

> > ### Author Response · Authors · 2018-11-27
> > **This is an interesting line of research, and it would be impactful to get this answer right!**
> >
> > Dear Ian,
> >
> > We also like to thank you for the time you dedicated and kindly read the revised version of our paper. As you mentioned, upon you and our four reviewers’ thoughtful comments, we improved the clarity of the presentation. We believe that the merit of openreview helps authors to deliver polished and influential research contributions. With this regard, we would be grateful to you if you could leave a comment for our AnonReviewer1 regarding drop-out and its correspondence to Thompson sampling. We already referred our AnonReviewer1 to the discussion in Appendix A of your BootstrapDQN paper. Moreover, we made an additional empirical study on four Atari games to show the deficiency of dropout in providing reasonable exploration and exploitation tradeoff. We would appreciate it if you could take a time and leave a comment in the corresponding thread.
> >
> > Regarding the confidence set C_t: The confidence C_t is mentioned in section 4.
> >
> > Regarding the discount factor: Both theorem 1 and theorem 2 hold for any discount factor 0<= \gamma<=1. We get a tight bound if we replace \sqrt(H) with a smaller quantity ||1,\gamma, gamma^2,...,\gamma^(H-1)||_2. We addressed this in terms of a remark in the latest version.
> >
> > Regarding the choices of prior/likelihood: We apologize that the choices of prior and likelihood were not clear from the main text. We would like to restate that we do not specify the choices of prior and likelihood. They can be anything as long as they satisfy the set of assumptions on page 6, e.g., sub-Gaussianity.
> >
> > Regarding the inconsistency: Conditioned on any data history H_t, the posterior over w is identical to the posterior of the optimal parameter. Similar theoretical justification is also deployed in Russo et al. 2014, “Learning to Optimize Via Posterior Sampling” page 10, as well as many of your papers, e.g., “(More) Efficient Reinforcement Learning via Posterior Sampling” lemma 1.
> >
> > Regarding your statement on “this is not a model-free algorithm if we know the likelihood”: Theoretically, given a w, the knowledge of the likelihood does not determine a model. These algorithms, also neither require to construct a model nor to store any MDP model parameters (e.g., transition kernel).
> >
> > In model-based PSRL, we generally specify the problem with a prior over MDP models, as well as a likelihood over state transitions and reward processes. These quantities are given and known in the model-based PSRL framework. Consequently, in model-free PSRL we specify the problem with a prior over Q and likelihood over the return where similarly these quantities are given and known.
> >
> > We agree with you that when the prior and the likelihood functions are arbitrary, then computing the posterior, as well as sampling from it, can be computationally hard. As you know, this is a principled issue with Bayesian methods. It is also an unsolved issue for model-based methods, e.g., in continuous MDPs. While we are excited about this line of research, we left the study of relaxing this computation complexity for the future work.
> >
> > We would like to thank you again for taking the time to leave thoughtful comments and we appreciate your positive assessment of this line of research. We would be also grateful to you if you could leave a comment on our AnonReviewer1 review regarding the drop-out discussion.
> >
> > Sincerely,
> > Authors

---

> > > ### Public Comment · ~Ian_Osband1 · 2018-11-28
> > > **Making progress, but still serious problems**
> > >
> > > This issue of "exact" posterior inference is crucial.
> > > Yes, in previous papers analysing PSRL they assume exact posterior inference.
> > > However, those papers do not claim that PSRL is a model-free algorithm when doing this.
> > >
> > > At this point, the analysis of this "model-free" PSRL is quite divorced from your algorithm Bayesian DQN (which is the model-free RLSVI, but only applying randomization to the final layer of a DQN).
> > > I think it would be better to separate these contributions into two separate papers...
> > >
> > > At the moment, there is an implication that the two algorithms (model-free PSRL and BDQN) are conflated... but this is confusing because actually they are not the same thing.
> > > It's also not clear if there are any examples where you can do this "model-free PSRL" exactly, even with infinite compute, without an underlying model of the MDP?
> > > The issue is that, in order to have the *exact* prior/likelihood updates, you need to take into account the max_a dynamics.
> > >
> > > I also think you should omit the "gamma" stuff entirely.
> > > Regret bounds O(\sqrt{T}) make absolutely no sense in a discounted setting... for any discount < 1 the regret is bounded O(1 / (1-gamma) ).

---

> > > > ### Author Response · Authors · 2018-11-28
> > > > **Posterior Sampling RL**
> > > >
> > > > Dear Ian,
> > > >
> > > > We would like to thank you for kindly leaving a comment on our AnonReviewer1's thread.
> > > >
> > > > Regarding the analysis of model-free PSRL; BDQN up to some modification reduces to PSRL algorithm if the prior and posterior are Gaussian. As we mentioned in the paper, since maintaining the posterior distribution can be computationally intractable, in our empirical study, we approximate the posterior with a Gaussian distribution. We apologize if it is not clear from the paper. We will emphasize more on this statement.
> > > >
> > > > As we mentioned before, in model-based PSRL, we generally specify the problem with a prior over MDP models, as well as a likelihood over state transitions and reward processes. As you also agreed, we are given these quantities before interacting with the environment. Consequently, in model-free PSRL, we specify the problem with a prior over Q and likelihood over the return. Similarly, we are given these quantities before interacting with the environment to pursue PSRL
> > > >
> > > > Regarding the discount factor \gamma;  the per episode regret is bounded above with O(1 / (1-gamma) ),   but not the overall regret. Following your statement, the regret is upper bounded as  O(T / (1-gamma) ) which is linear in T. We derived a sublinear regret of \tild{O}(\sqrt{T} ||(1,\gamma,\gamma^2,...,\gamma^{H-1})||_2 ).
> > > >
> > > >
> > > > We would like to appreciate the thoughtful comments on our paper and also thank you for taking the time to leave a comment on our AnonReviewer1 review regarding the drop-out discussion.
> > > >
> > > > Sincerely
> > > > Authors

---

### Public Comment · ~Akshay_Krishnamurthy1 · 2018-12-07
**About Assumption 1**

Hi authors,

I was studying Appendix B, which contains the main theoretical content, and I have a question about Assumption 1. Above equation (5), there is a comment that says "the noise process is correlated, dependent on the agent policy, and is not mean zero unless under policy \pi^\star." Then, Equation 7 is just a re-writing of Equation 5 (there is a typo here since it should be \omega^\star). But Assumption 1 now states that \bar{\nu}_t is mean zero, which implies \nu_t is mean zero as well (by linearity of expectation).

These two statements are in contradiction, but Assumption 1 seems incorrect to me. One cannot model the random return for a suboptimal policy as an unbiased estimate of Q^\star, which is what Equation 7 + Assumption 1 are saying. If one executes a suboptimal policy, the random return will systematically underestimate Q^\star. This systematic underestimation/bias is the core difficulty in analyzing algorithms for linear Q-learning. So one cannot assume it away.

Could you please clarify?
Akshay

---

> ### Author Response · Authors · 2018-12-13
> **bias**
>
> Dear Akshay
>
> I really appreciate this comment. I could not find any explanation of how I missed this obvious point. As you can imagine, under this assumption, the problem becomes trivial. Let me relax this assumption ( which results in bias estimators for all the time step except for time step H) and see how the bound changes. Thanks again.
>
> Cheers

---

> > ### Author Response · Authors · 2018-12-14
> > **update**
> >
> > Dear Akshay
> >
> > Thanks again for bringing up this important point.
> >
> > Update:
> >
> > For the general case, with a modification in the current proof, I get a regret upper bound of
> > d^((H+1)/2)\sqrt(T).
> >
> > With an additional prior of
> > sum_{i<=t}||\phi(x_i,\pi_t(x_i))||^2_{\xi_t^-1}=O(1)
> > I get d\sqrt(T).
> >
> > I'll update the text with this regard.
> >
> > Cheers

---

> > > ### Public Comment · ~Akshay_Krishnamurthy1 · 2018-12-17
> > > **Thanks, but this much weaker**
> > >
> > > Hi authors,
> > >
> > > Thanks for looking into this. The d^{(H+1)/2}\sqrt{T} regret bound is less surprising to me, but this is not a very good guarantee. In particular, I think an exp(d) regret guarantee does not provide evidence for sample efficiency.
> > >
> > > About the "prior" (I guess you mean assumption?). I do not think it is reasonable to make this assumption, as it depends on the policy \pi_t, which of course is algorithm dependent. Are there any special cases where this assumption is true, in some algorithm-independent sense?
> > >
> > > Akshay

---

> > > > ### Author Response · Authors · 2018-12-18
> > > > **the assumption**
> > > >
> > > > Dear Akshay
> > > >
> > > > I totally agree with you that the d^{{H+1}/2} dependent regret upper bound does not evidence the sampling efficiency, and I also agree it is not reasonable to just assume the mentioned assumption is satisfied.
> > > >
> > > > I am currently working on the cases that this assumption is rigorously satisfied (e.g. following optimism under what condition this is satisfied). So far, the current stage of the proof, O(d\sqrt(T)) with the independence assumption, is trivial and moreover the upper bound O(d^{{H+1}/2} \sqrt(T)) is not tight, therefore the current state of the theoretical contribution is not complete yet. I would like to thank you again for your thoughtful and helpful comment on this paper.
> > > >
> > > > Cheers

---

### Meta-Review · Area_Chair1 · 2018-12-13
**A neat idea with impressive results but has technical flaws and issues with clarity**

**Confidence:** 5
**Recommendation:** Reject

**Metareview:**

There was a significant amount of discussion on this paper, both from the reviewers and from unsolicited feedback.  This is a good sign as it demonstrates interest in the work.  Improving exploration in Deep Q-learning through Thompson sampling using uncertainty from the model seems sensible and the empirical results on Atari seem quite impressive.  However, the reviewers and others argued that there were technical flaws in the work, particularly in the proofs.  Also, reviewers noted that clarity of the paper was a significant issue, even more so than a previous submission.

One reviewer noted that the authors had significantly improved the paper throughout the discussion phase.  However, ultimately all reviewers agreed that the paper was not quite ready for acceptance.  It seems that the paper could still use some significant editing and careful exposition and justification of the technical content.

Note, one of the reviews was disregarded due to incorrectness and a fourth reviewer was brought in.